# The Preparation, Structural Design, and Application of Electroactive Poly(vinylidene fluoride)-Based Materials for Wearable Sensors and Human Energy Harvesters

**DOI:** 10.3390/polym15132766

**Published:** 2023-06-21

**Authors:** Weiran Zhang, Guohua Wu, Hailan Zeng, Ziyu Li, Wei Wu, Haiyun Jiang, Weili Zhang, Ruomei Wu, Yiyang Huang, Zhiyong Lei

**Affiliations:** 1School of Packaging and Materials Engineering, Hunan University of Technology, Zhuzhou 412007, China; jetwalkerz@gmail.com (W.Z.); wghaiyyr1120@163.com (G.W.); zenghl0424@163.com (H.Z.); ziyv.lee@gmail.com (Z.L.); waqarrd@163.com (W.W.); zh_weili@163.com (W.Z.); cailiaodian2004@126.com (R.W.); 2National & Local Joint Engineering Research Center for Advanced Packaging Material and Technology, Hunan University of Technology, Zhuzhou 412007, China; 3Shenzhen Glareway Technology Co., Ltd., Shenzhen 518110, China; kshjl@126.com (Y.H.); leizhiyong468@126.com (Z.L.)

**Keywords:** poly(vinylidene fluoride), wearable sensor, nanogenerator, energy harvester, electroactive, piezoelectricity, triboelectricity

## Abstract

Owing to their biocompatibility, chemical stability, film-forming ability, cost-effectiveness, and excellent electroactive properties, poly(vinylidene fluoride) (PVDF) and PVDF-based polymers are widely used in sensors, actuators, energy harvesters, etc. In this review, the recent research progress on the PVDF phase structures and identification of different phases is outlined. Several approaches for obtaining the electroactive phase of PVDF and preparing PVDF-based nanocomposites are described. Furthermore, the potential applications of these materials in wearable sensors and human energy harvesters are discussed. Finally, some challenges and perspectives for improving the properties and boosting the applications of these materials are presented.

## 1. Introduction

The development and utilization of renewable, sustainable, and environmentally friendly energy sources are essential to mitigate the continuously rising global energy demand, shortage of fossil fuels, and environmental pollution caused by non-renewable sources [1,2,3]. To this end, various energy harvesting, storage, and recycling technologies based on external sources (e.g., solar power, thermal energy, and chemical energy) have been developed. Among them, mechanical energy sources are readily available in nature and daily human activities, such as human movement with fingers, hands, arms, legs, etc., speaking, respiration, airflow, vibrations, frictional forces, water precipitation, and hydraulics (waves in nature, blood flow inside organisms, etc.) [4,5,6,7]. Over recent years, self-powered wearable sensors and human energy harvesters based on nanogenerators (NGs) have attracted considerable attention, including piezoelectric nanogenerators (PENGs) and triboelectric nanogenerators (TENGs). These wearable sensors can be used to detect, monitor, and record real-time information on the human physiological status.

Poly(vinylidene fluoride) (PVDF) is one of the most interesting semicrystalline polymers and is often used in sensors, actuators, energy harvesters, etc., because of its high biocompatibility, film-forming ability, low cost, excellent chemical stability, and good electroactive characteristics, including piezo-, pyro-, and ferro-electric properties [8,9,10,11]. Notably, PVDF-based NGs can effectively harvest energy from organic systems and human activities, such as body motion and even breathing [12,13,14,15]. In addition, the excellent biocompatibility of PVDF-based polymers makes them desirable for application in flexible membranes, energy sensors, energy-harvesting electronic skins (e-skins), and even implantable devices and artificial prosthetics [16]. However, they still have some drawbacks, such as low ionic conductivity, low crystallinity, and shortage of reactive groups [17]. The low crystallinity especially can limit their piezoelectric properties, charge mobility, and dielectric constant.

Moreover, two copolymers of PVDF are popular candidates for self-powered electronics and energy harvesting. The first one, poly(vinylidene fluoride-co-trifluoroethylene) (P(VDF-TrFE)), is promising due to its thermodynamic stability and high crystallinity. TrFE has a more rigid and ordered structure compared to the vinylidene fluoride (VDF) monomer in PVDF. This structural difference promotes the formation of crystalline regions within the copolymer, where it allows for efficient alignment of polymer chains, resulting in enhanced charge generation in response to mechanical stress or strain. However, the high cost, poor thermal stability, limited stacking integrity, chemical reactivity, and poor ferroelectric dipole density of P(VDF-TrFE) restrict its large-scale device fabrication. Although the copolymerization units of TrFE can improve the crystallinity of PVDF, their crystal defects often cause current leakage paths [18,19]. Compared with P(VDF-TrFE), PVDF homopolymers exhibit a higher dipole density and thermal stability. The other PVDF copolymer, poly(vinylidene fluoride-co-hexafluoropropylene) (P(VDF-HFP)), has a relatively higher piezoelectric sensitivity and electrostrictive strain [20,21,22,23]. Furthermore, the piezoelectric coefficient of P(VDF-HFP) (13.5 with 5% HFP) is much higher than those of PVDF (≈12.9) and P(VDF-TrFE) (≈10.4) copolymers [24]. Furthermore, the P(VDF-HFP) copolymer has a unique piezoelectric response, which makes it more suitable for fabricating self-powered wearable, stretchable electronic devices, compared with the other polymers. The PVDF- and PVDF-HFP-based materials exhibit immense potential as electrolytes in solid-state lithium-ion batteries, owing to their large dielectric constant, chemical stability, and high mechanical strength [25,26].

Apart from PVDF co-polymers (e.g., P(VDF-TrFE) and P(VDF-HFP)), other polymers such as polypropylene (PP), Nylon11, polylactic acid (PLLA), and poly (lactic-co-glycolic acid) (PLGA) also exhibit piezoelectric properties [27]. Moreover, their soft property is suitable for wearable electronics. Although several traditional ceramic materials such as lead zirconate titanate (PZT) can efficiently convert mechanical energy into electrical energy, they are rigid and difficult to manipulate and machine. On the other hand, owing to its flexibility, PVDF shows excellent long-term stability and does not depolarize when exposed to extremely strong alternating electric fields. Consequently, PVDF-based flexible films have gained immense popularity in recent years.

In this review, we focus first on the structure and identification of various PVDF phases. Then, various methodologies for obtaining electroactive phases are discussed, including phase transformation techniques and direct fabrication methods. Next, the structural design methods and the applications of PVDF structures in wearable sensors and human energy harvesting are outlined. Finally, some existing challenges are highlighted, and prospective solutions are suggested for boosting the development and application of PVDF-based sensors and energy harvesters.

## 2. PVDF Phase Structure and Identification

### 2.1. PVDF Phase Structure

It has been widely established that the semicrystalline PVDF polymer shows five distinct crystalline phases: the α-, β-, γ-, δ-, and ε-phases [28,29,30,31,32,33,34,35,36,37], which have different stereochemical macromolecular conformations. Firstly, the α-phase, the most thermodynamically stable polymorph, is a non-electroactive, nonpolar, and paraelectric phase with no piezoelectricity, and has a centrosymmetric (P2_1_/c) monoclinic unit cell with alternating trans and gauge linkage (TGTG’) conformation [38,39]. On the other hand, the β-phase, the most electroactive phase with excellent piezoelectricity, has an orthorhombic crystal structure with all trans (TTT) planar zigzag conformation [40]. In PVDF, the electroactive β-phase is the most preferred, due to its superior piezo-, pyro-, and ferro-electric performance [41]. Usually, it is important to transform the α-phase into β- phase, because the α-phase is the major component of PVDF films [42]. High isothermal crystallization temperatures often result in the formation of the γ-phase, which also possesses an orthorhombic crystal structure with a T_3_GT_3_G’ conformation [33,34,35]. The δ- and ε-phases are the polar and antipolar analogues of the α- and γ-phases, respectively [31,32,36,43]. Compared with the α-phase, the δ-phase has a non-centrosymmetric (P2_1_cn) unit cell, rendering it piezoelectric, pyroelectric, and ferroelectric [44]. Similar to the β-phase, the δ-phase has superior memory functionality [45,46]. Therefore, the δ-phase is a promising alternative to the β-phase in PENGs. Among the five phases, the α-, β,- and γ-phases are the most widely investigated. Their chain conformations are shown in Figure 1 [47].

Furthermore, the electromechanical coupling factor, *k*, is one of the dominate parameters for the preparation and application of PVDF-based materials. It presents the efficiency in the mechanical to electrical transformation. High crystallinity and preferred orientation in PVDF crystallites can lead to high remnant polarization, which increases the electromechanical coupling factor. PVDF with different phases have distinct electromechanical coupling factors, and they are also influenced by temperature, poling condition, etc. [47]. Another significant parameter, *d_33_*, is used to represent the piezoelectric constant in PVDF-based materials, which often has a negative sign conversion resulting from the crystal structure and molecular alignment. It signifies that the resulting electric field is in the opposite direction to the applied stress or strain.

### 2.2. PVDF Phase Identification

Although the β- and γ-phase have a similar conformation, the piezoelectric effect of the β-phase is stronger than that of the γ-phase. Therefore, effective strategies for obtaining the electroactive phase of PVDF have garnered considerable research attention, and the identification of the α-, β-, and γ-phases is a crucial step in realizing this goal. Among the identification approaches, Fourier-transform infrared (FTIR) spectroscopy and X-ray diffraction (XRD) are considered to be the most reliable ones. Usually, both these techniques are simultaneously used to better discern the β- and γ-phases.

It has been reported that the α-, β-, γ-, and δ-phases have distinct characteristic bands in the FTIR spectrum (Table 1). The intensity of these bands indicates the orientation of the CF_2_ dipole moment. For example, the broad spectral band at 840 cm^−1^ results from the overlap of the β- and γ-phases, so other identification approaches must be applied to discriminate between the two phases.

The relative proportion of electroactive phases (*F_EA_*) can be utilized to distinguish some phases. For example, taking the band at 840 cm^−1^ as an example, S. Maji et al. [61] deconvoluted the FTIR spectrum (900–750 cm^−1^ bands) and quantified the relative fraction of electroactive phases (*F_EA_*), including both β- and γ-phases, using the following equation:(1)$$F_{EA}=\frac{I_{EA}}{\left(\frac{K_{840}}{K_{763}}\right)I_{763}+I_{EA}}\times 100$$
where *F_EA_* represents the proportion of the electroactive phase; *I*_763_ and *I*_EA_ are the absorption intensities at 763 and 840 cm^−1^, respectively; and *K*_763_ and *K*_840_ are the absorption coefficients at the respective wavenumbers [62]. The individual β- and γ-phases of PVDF films can also be defined by curve deconvolution of the band at 840 cm^−1^. The ratio of the electroactive β- and γ-phases can be obtained as follows [63]:(2)$$F\left(\beta \right)=F_{EA}\times \left(\frac{A_{\beta }}{A_{\beta }+A_{\gamma }}\right)\times 100\%$$
and
(3)$$F\left(\gamma \right)=F_{EA}\times \left(\frac{A_{\gamma }}{A_{\beta }+A_{\gamma }}\right)\times 100\%$$
where *A*_β_ and *A*_γ_ are the total regions under the deconvoluted curves of the β and γ-phases centered at the 840 cm^−1^ band.

It has been widely accepted that the absorption band at 840 cm^−1^ is common to both β and γ-phases, but it exists as a strong band only for the β-phase, while it appears as a shoulder of the 833 cm^−1^ band for the γ-phase.

It can be seen in Table 1 that some peaks of the α-, β- and γ-phases overlap with each other, so it is difficult to distinguish them just by FTIR spectroscopy. XRD characterization is another auxiliary approach to discriminate the structures. The representative crystal diffraction planes and diffraction angles of the various phases of PVDF are listed in Table 2. The peak at 20.6° is attributed to the (110) and (200) crystal planes of the β-phase, while the peaks at 18.5°, 19.2°, and 20.4° correspond to the (020), (002), and (110) crystal planes of the γ-phase. Although both the α- and δ-phases have similar chain conformations, the intensities of peaks at 2θ = 17.6° and 25.6° corresponding to (100) and (120) planes are different for the two phases. After heat treatment at 170 °C, the lattice shape and size as well as the symmetry of the unit cell lattice are changed [64]. However, some peaks cannot be easily distinguished. For example, the characteristic peak of the α-phase at 18.3° (020) is often overlapped with that of the γ-phase at 18.5° (020); the broad peak at 20.5° often results from the overlap of the β-phase signal at 20.6° and the γ-phase signal at 20.4°. Sometimes a broad double peak appears around 20.4°, indicating the coexistence of β- and γ-phases [37].

The overall crystallinity (*X_c_*) is calculated according to the crystalline and amorphous regions isolated from the XRD patterns by the Gaussian function, as follows:(4)$$\;X_{c}=\frac{\sum A_{cr}}{\sum A_{cr}+\sum A_{amr}}\times 100\%$$
where ∑*A_cr_* and ∑*A_amr_* are the sums of integrated areas from crystal diffraction peaks and the amorphous halo. The crystallite size can be determined using the Debye–Scherrer formula, as follows:(5)$$t=\frac{\lambda }{B\cos\theta }$$
where t is the crystallite size, *B* is the FWHM of the diffraction peak in radians, and λ is the X-ray wavelength.

Other auxiliary approaches can be implemented, based on the physical properties of PVDF. For example, the melting temperature of the α-phase is lower than that of the polar β- and γ-phases in PVDF, so differential scanning calorimetry (DSC) is suitable for identifying the α-phase in relation to the β- and γ-phases [69,70].

## 3. Phase Transformation Methodologies

β-, γ-, and δ- phase PVDF show piezoelectric, pyroelectric, and ferroelectric properties, which make them good candidates for obtaining electroactive films. Phase transformation and suppression of the non-electroactive phase are useful for obtaining these electroactive phases. Polarization treatment at high temperature and high electric field [71,72], corona poling [73], and mechanical stretching [9,71,74] have been confirmed as effective phase transformation techniques. Recently, self-polarization of oriented β-phase crystallites was reported in ultrathin PVDF-based films synthesized using spin-coating [75] and the Langmuir–Blodgett (LB) [76,77] technique. This phenomenon was attributed to the built-in electric field [75], in-film stress, and the strong hydrogen bonding interaction between PVDF molecules and water [76,77]. In these cases, mechanical stretching and tension can result in enhanced piezoelectric effect.

Numerous efforts have been made to apply electroactive β- or γ-phase PVDF in devices such as sensors and energy harvesters. However, only a few studies have focused on δ-phase PVDF because it requires multi-step processing and an ultrahigh-intensity electric field (~170 MV/m). Moreover, it is often difficult to measure the piezoelectric activity unless an electric field is applied to the PVDF layer, which aligns the anisotropic molecular dipoles of the CH_2_/CF_2_ along the electric field direction. Over the years, to avoid the usual poling stages, nanoparticle (NP) doping, the introduction of hydrated salt film, and the Langmuir-Schaefer (LS) method have been utilized to replace electric field poling in β-phase PVDF films.

### 3.1. Mechanical and Temperature Control

Controlling the mechanical properties and temperature is typically the first method utilized to obtain the electroactive phase (β-, γ-, or δ-phase) of PVDF, because mechanical stretching or heat treatment can cause recrystallization [78]. Mechanical control is usually achieved by pressure, stretching, milling, solvent evaporation, and so on. Among these, pressure is commonly used in combination with temperature. Doll and Lando demonstrated that if PVDF undergoes heat treatment at 280 °C and pressure crystallization at 5000 atm, the electroactive β phase can be obtained [79]. This may be attributed to the formation of a high-density phase under pressure crystallization. When the PVDF film is subjected to high temperature and pressure, the α-phase with a relatively lower density is transformed into a high-density β-phase. In fact, γ-phase PVDF can form the α-phase or melt under high-pressure heat treatment.

Based on temperature control, many heat treatment techniques are employed to control the formation of the β- and -γ phase, including quenching [80,81,82], annealing [83,84,85], and low supercooling [62,86]. Moreover, there is a transformation from the α- to δ-phase under heat treatment at a particular temperature (170 °C). W. M. Prest Jr. and D. J. Luca suggested that there is a temperature range in which both the α- and β- phases of PVDF can be formed at the same time [87]. A similar crystal transformation was observed by employing a position-sensitive proportional counter, an X-ray system [88], and micro-differential thermal analysis [81,89]. T. Hattori et al. reported that an elevated temperature can sometimes eliminate structural defects [90].

Mechanical stretching is the most popular method for the mass production of α-phase PVDF, which involves the successive melting and cooling of PVDF film [91]. The α-phase samples are stretched at high temperatures, which causes a decrease in the elastic modulus. During the stretching process, the thickness is reduced, and the α-to-β phase transformation begins with necking [92], which can be further improved by a poling field. Furthermore, during stretching, the crystallites transform from a spherulitic to a micro-fibrillar structure.

Y. A. Huang et al. proposed a novel helix electrohydrodynamic printing method in combination with in-surface self-organized buckling to prepare aligned PVDF nano/micro-fibers with excellent piezoelectric properties, using in situ mechanical stretching and electrical poling (Figure 2a,b) [93].

In PVDF films, the α-phase can be directly obtained during crystallization from the melt [74]. However, the β-phase films are obtained by drying the spin-coated films from 30 to 60 °C [73]. They are fabricated by rapidly quenching the films into a nonsolvent bath for inducing liquid–liquid and liquid–solid phase separation [94,95,96]. Several parameters such as composition [73,94], solvent type [97,98], quenching temperature, etc. [62,99] determine the crystalline phase and film microstructure.

Furthermore, β-phase PVDF can be obtained from α-or γ-phase PVDF using mechanical deformation [100], and the β-phase has an all-trans zigzag chain conformation, which can be observed under alternate changes in electric field direction [101]. S. J. Kang et al. presented a micropatterning approach to prepare patterned arrays of isolated ferroelectric γ-phase regions, which were embedded in the nonpolar α-phase in thin PVDF films [102]. With this process, only certain areas compressed by a patterned poly(dimethylsiloxane) (PDMS) mold were converted into a polar γ-phase structure. D. M. Esterly and B. J. Lov used cryogenic mechanical milling to successfully convert α-phase PVDF powder into β-phase powder [65]. It was also reported that further milling of the powders may reduce the crystallinity.

Apart from cryogenic mechanical milling and pressure crystallization, stretching is another important process for obtaining the electroactive phase. This process is usually combined with other procedures such as electric field poling, doping, etc. Shearing and stretching the PVDF molecular chains during the spin-coating process can lead to preferential formation of the β-phase. The crystal structure of the PVDF films is governed by many parameters, including the stability of the molecular chains caused by spin-coating and the evaporation rate of the solvent [103]. It has also been reported that the crystalline phase of the films is greatly affected by the dissolution of PVDF in different solvents. For instance, the well-dissolved PVDF solution can easily form the α-phase during the phase inversion process. Conversely, PVDF solution with poor dissolution is beneficial for forming the β-phase [104].

Ultrasonic treatment is another mechanical stretching technique, which is based on stress-induced effects driven by ultrasound energy. This process has been proven to increase the proportion of electroactive β-phase PVDF in composites, leading to a relatively high self-polarization performance compared with other external electrical poling methods [47]. Remarkably, highly efficient poly(vinylidene fluoride)-activated carbon (CPVDF-AC) composite films can be fabricated using the sonication method [105] without an additional electrical poling technique. During the ultrasonication process, the non-piezoelectric α-phase of PVDF can be transformed into the highly electroactive β-phase by using suitable solvents (dimethylformamide (DMF), acetone, etc.). The sound energy breaks the intermolecular interactions and vibrates the -CH_2_-/-CF_2_- electric dipoles of the PVDF to obtain a homogeneous transparent DMF solution. The activated carbon (AC) filler plays a crucial role as a stabilizer for improving the electroactivity of the β-phase, and also provides an electrical conduction path between the -CH_2_-/-CF_2_- electric dipoles of the PVDF. The instantaneous electrical density of the NG has been reported to be around 63.07 mW/m^2^, which is not high but is sufficient for application in low-energy electronics such as light-emitting diodes (LEDs) and displays.

### 3.2. Electric Field Poling

Electric field poling, including corona and plasma poling, is a common strategy for forming β-phase PVDF. This technique can also avoid electric breakdown without heat treatment at voltages up to ±10 kV. P. D. Southgate demonstrated a complete poling of PVDF film in less than 1 sec by corona charging at room temperature and about 2 °C [106]. This process was accompanied by a transformation from the α- to the β-phase and an enhancement in the pyroelectric coefficient. D. K. Das-Gupta and K. Doughty also reported that corona charging of PVDF at normal temperatures may be a more appropriate poling method than the conventional high-temperature poling method to obtain comparable values of the piezoelectric coefficient [107]. G. T. Davis et al. further confirmed that at room temperature, the threshold electric field value of 1 MV/cm can promote the transformation from α- to β-phase [108]. Y. Jung et al. conducted the corona poling process on a PVDF film to improve its piezoelectricity at 80 °C for 20 min under 6.5 kV. The value of the piezoelectric constant d_33_ was found to be around −3.5 pC/N [109]. This film was used to fabricate a piezoelectric artificial cochlea (PAC) device, which realized the frequency separation of the incoming mechanical signal from a micro-actuator into a frequency bandwidth within the 0.4–5 kHz range. J. E. McKinney et al. used the plasma poling technique to polarize a PVDF film [110]. A permanent polarization transformation from the α- to β-phase can be achieved by the application of DC electric fields greater than 1 MV/cm for a few seconds. The α-phase can transform to the δ-phase under an electric field (i.e., the polar form II or IV), and a higher electric field ultimately causes the transformation from the δ- to β-phase.

However, electric field poling is typically not preferable because it consumes a large amount of electricity and is prone to electric breakdown failure. Furthermore, it is impractical for specialized applications where unique patterns are desired, such as e-skins, robotic interface, etc. [53,111].

### 3.3. Adding Fillers

Some fillers are applied as nucleating agents to induce the β- or γ-phase transformation of PVDF. Usually, these nucleating agents form hydrogen bonds with -CF_2_ and align the dipoles.

The incorporation of metal oxides has been proven to be useful in realizing the phase transformation of PVDF. M. Alam et al. showed that TiO_2_ NPs enhanced the piezoelectric β-phase content and mechanical properties of composite PVDF nanofiber [112]. However, the piezoelectric PVDF matrix decomposed over time, due to the photoactivity of α-TiO_2_ NPs, lowering the device performance. S. H. Kim et al. prepared F-coated rutile TiO_2_ NPs and demonstrated that the F-coated rutile-TiO_2_ NP-doped composite film efficiently induced the piezoelectric phase transition of non-electroactive PVDF, due to the highly electronegative F bonds on the surface of these NPs. Under intense sunlight and 2.0 wt.% composite film, 99.20% of the non-electroactive PVDF was converted to the electroactive phase. Furthermore, the F-coated rutile-TiO_2_ NPs were used to prepare a piezoelectric device which exhibited excellent piezoelectric performance and durability under 64 h of photoirradiation at an intensity of 0.1 W/cm^2^ [113].

Inducing spontaneous polarization is another promising strategy for obtaining the electroactive phase of PVDF. For example, S. Garain et al. doped cerium(III)-N,N-dimethylformamide-bisulfate [Ce(DMF)(HSO_4_)_3_] complex into PVDF to increase the yield of the electroactive β- and γ-phases to 99% (Figure 3a,b) [114]. This transformation is mainly attributed to the electrostatic interactions between the fluoride ions in the PVDF and the surface-active cation clusters of cerium through hydrogen bonding and/or bipolar forces between the opposite poles of the cerium complex and the PVDF (Figure 3c,d). This PVDF composite film was used to fabricate n NG for harvesting energy from simple repeated human finger imparting. A remarkable output voltage (~32 V) could be obtained with an additional ultraviolet (UV) light-emitting capability. Q. Li et al. fabricated composite films with a high content of β-phase in the PVDF by combining PVDF and a molecular ferroelectric filler with multiple polarization axes (dabcoHReO_4_) [115]. An evident phase transformation from α- to β-phase in the PVDF was obtained by examining the hydrogen bond interaction between the PVDF matrix and the dabcoHReO_4_ filler.

### 3.4. Other Methods

Some other methods have been employed to realize the electroactive phase transformation in PVDF. For example, S. Maji et al. demonstrated ferroelectric switching and piezoelectric behavior in an ultrathin PVDF film fabricated using the horizontal LS method [61]. The pure β-phase was obtained just by increasing the number of LS layers, without requiring any additional non-ferroelectric agents. The edge-on oriented CH_2_–CF_2_ units of PVDF at the air–water interface facilitated the self-orientation of the ferroelectric dipoles through the hydrogen bonding network. M. K. Lee and J. Lee used a nano-frost array technique to prepare a γ-phase PVDF porous membrane [67]. This method facilitated the arrangement of nano-frost crystals on the PVDF surface. Furthermore, the freezing technique caused an improvement in the crystallinity and polymorphic controllability. Notably, it was crucial to choose a suitable solvent for the PVDF film.

Besides the transformation, stabilization of the electroactive phases is also significant for improving performance in piezoelectric materials. Whiter et al. reported the influence of confinement on the crystallization behavior and phase transition of PVDF by utilizing nanostructured templates at room temperature [116]. The research shows that confinement improves the alignment and structure of PVDF chains, resulting in enhanced piezoelectric properties.

## 4. Direct Methods for Preparing Electroactive Phases

### 4.1. Electrospinning

Electrospinning is widely established as a simple, economical, and versatile technology for fabricating triboelectric nanogenerators (TENGs) [117], sensors [118,119,120], piezoelectric nanogenerators (PENGs) [121,122], and photoelectric scanners (PESs) driven by magnetic force [112,123,124,125,126,127]. Over recent years, electrospun fiber geometries have been used to develop different kinds of PVDF structures, such as rods, tubes, particles, and flakes [128,129]. In contrast to the nanofiber membranes prepared by the conventional method, electrospun membranes have higher flux, better effective porosity, higher selectivity, and lower cost [130].

The electrospinning technique includes three fundamental components: a powerful electric field to charge a needle tip and solution/melted mixture droplets, a steel needle with a tiny tip, and a grounded collecting screen [131]. The electrostatic force opposes the surface tension of the droplets, so they burst and emit a jet stream of nano-scale fibers.

Electrospinning is sometimes combined with stretching, doping, and melt recrystallization for obtaining a higher content of the electroactive phase [43,110,132,133,134]. Over the recent years, spinning has attracted much attention, due to two main reasons. Firstly, the fabrication and poling of films are finished at the same time. Secondly, the films prepared by the spinning technique are very flexible and are suitable for use in wearable and stretchable electronics. D. Mandal et al. used a single-stage electrospinning process to obtain preferentially oriented induced dipoles in P(VDF-TrFE) nanofibers [135]. The as-electrospun P(VDF-TrFE) nanofiber networks could be used as flexible NGs and nano-pressure sensors. To obtain a higher fraction of the β-phase, many other poling strategies have been applied simultaneously. Y. Huang et al. proposed a kinetically controlled mechanoelectrospinning (MES) technique to directly write diversified hierarchical micro/nanofibers in a continuous and programmable manner [136]. Compared with the conventional electrospinning technique, the MES process introduced a mechanical drawing force to provide advantages such as tunable and high resolution, wide adaptability to inks with different viscosities, simultaneous control of the location and morphology of the formed structures, and immediate deposition of smooth layered structures. Similarly, Duan et al. obtained non-wrinkled, highly stretchable piezoelectric structures on a prestrained PDMS substrate by electrohydrodynamic direct-writing [137]. Using PVDF nanofibers, Y. Ding et al. fabricated an energy harvester using the mechano-electrospinning process (Figure 4a–c) [138], which could generate electricity under bending, stretching, and compression. Lee et al. proposed the electric poling-assisted additive manufacturing (EPAM) process to directly and continuously print piezoelectric devices from PVDF polymeric filament rods under a high electric field [139]. During the EPAM process, the molten PVDF polymer was simultaneously mechanically stressed in situ by the leading nozzle and electrically poled by applying a high electric field under high temperature. Furthermore, the PVDF polymer dipoles remained well aligned and uniform over a large area during the continuous fabrication process.

However, conventional electrospun nanofibers have some disadvantages. For example, they wrinkle, and their power transfer efficiency declines after being exposed to ambient air for a certain time. Moreover, compression is not effective at relatively high frequencies, so they might not be able to provide enough deformation for application in piezoelectric generators [141]. Like many other commercial nanofiber membranes, heavy and irreversible fouling remains a major limitation of the electrospun microfiltration membranes [142,143]. C. Dong et al. used PVDF/TiO_2_ nanofiber to prepare a flexible self-powering/self-cleaning e-skin using the high-voltage electrospinning method for actively detecting body motion and degrading organic pollutants. The photocatalytic activity of TiO_2_ and the piezoelectric effect of PVDF were coupled in a single physical/chemical process to efficiently degrade organic pollutants on the e-skin. For example, methylene blue (MB), could be totally degraded within 40 min under UV/ultrasonic irradiation [144]. Compared with electric field poling, electrospinning is more useful for imparting the piezoelectric property in PVDF while also degrading the organic pollutants [145].

Applying an external magnetic field is an effective auxiliary approach during the spin-coating process. S. M. Harstad et al. prepared TENGs using the combination of self-polarized, high β-phase nanocomposite films of Gd_5_Si_4_-PVDF and polyamide-6 (PA-6) films using the phase inversion method under a magnetic field, which generated a significantly higher voltage of 425 V, a short-circuit current density of 30 mA/m^2^, and a charge density of 116.7 μC/m^2^, as compared to the pristine PVDF-(PA-6) combination [146].

Near-field electrospinning (NFES) is widely used to fabricate one-dimensional (1D) and two-dimensional (2D) high-level aligned nano/micro-fibers from polymer solutions. For example, Y. K. Fuh et al. combined the NFES technique for PVDF micro/nanofibers and the three-dimensional (3D) printing method for a topologically tailored substrate to fabricate a wavy substrate self-powered sensor (WSS) [147]. This sensor exhibited a significantly enhanced piezoelectric output, which was attributed to the long fiber length. Furthermore, the integrated 3D structure could be directly utilized on a wearable device.

The piezoelectric properties of as-prepared membranes can be optimized by tuning the electrospinning parameters such as acetone percentage, distance between the tip and collector, flow rate, and voltage configuration. Dimethylformamide (DMF) is the most popular solvent used in this method, but dimethyl sulfoxide (DMSO) is also a promising solvent, which needs to be further investigated because of its relatively lower toxicity [148].

Recently, a novel spinning method known as the solution blow spinning (SBS) technique has been reported (Figure 4d), which is an alternative process for fabricating nanofibers by applying high-speed airflow. It has many benefits over electrostatic spinning, such as a more straightforward procedure, lower power requirements, and higher production efficiency [140].

### 4.2. Melt Spinning

The melt spinning method is a specific approach, in which insoluble polymers and ceramic materials are utilized in the electrospinning technology. Low loading of additive materials in the shape of nanorods is beneficial for imparting a better piezoelectric property. Additionally, the surface charge and size of the filler have a significant effect on the nucleation of the β-crystalline phase in the PVDF composite during this procedure [149]. Bairagi et al. used β-phase PVDF as a polymer matrix and various percentages of potassium sodium niobate (KNN) nanorods as a filler material to fabricate a PENG by melt spinning [150]. Compared with other piezoelectric ceramic materials, KNN has obvious advantages, such as higher piezoelectric coefficient, higher dielectric constant, higher electromechanical coupling factor, and environmental friendliness [151]. KNN is also a lead-free ceramic-based material, which possesses high-temperature stability and piezoelectric properties, simultaneously.

### 4.3. Blending

Several researchers have combined PVDF with other materials to enhance the mechanical properties, electroactivity, flexibility, and other performance parameters of PVDF-based composite materials.

Graphene [152] and graphene oxide (GO) [153,154] are common modifiers for improving the output, charging capability, durability, and crystallinity of the electroactive phase (β/γ phase). Fe-doped reduced graphene oxide (rGO) nanosheets can induce the conversion from α- to γ-phase by ion–dipole and/or hydrogen bonding interactions [58]. At the same time, the addition of Fe-doped rGO can enhance the electrical energy density to around 0.84 J·cm^−3^ under an electric field of 537 kV·cm^−1^. A high stable yield (99%) of γ-phase PVDF can be obtained by controlling the fraction of Fe-doped rGO. AlO-rGO also acts as a nucleating agent for electroactive β-phase formation. Hu et al. reported that the presence of rGO in PVDF-TrFE composites significantly improved the crystallinity of the β-phase PVDF-TrFE and enhanced the formation of hydrogen bonds via the interaction of dipoles between rGO and PVDF-TrFE [155]. Karan et al. used AlO-rGO as a nucleating agent to fabricate a hybrid piezoelectric nanogenerator (HPENG) with an excellent energy harvesting capacity [156].

Metal and metal oxides, including NiO [157], CuO [158], ZnO [159], and CoFe_2_O_4_ [160,161], are widely used to improve the β- or γ-crystalline phase fraction in PVDF composite materials [32,162,163,164]. For instance, ZnO-NPs can induce a complete γ-phase in PVDF, where conventional electrical poling is unnecessary for the generation of piezoelectric properties. Cheng et al. fabricated an integrated and self-powered UV sensor composed of PVDF and ZnO nanowire film [165]. The ZnO nanowires applied internal strain to the PVDF in the composite system, which increased the electrical power output of the hybrid NG [166]. The addition of tetrapod ZnO can also provide a motion-powered tactile-perception function and a self-clean property for piezoelectric PVDF composites [167]. Thus, this composite is a promising candidate material for e-skins to monitor body motions such as elbow bending or finger pressing and eliminate organic pollutants and bacteria. Moreover, the ZnO particles, which act as a nucleating agent to guarantee a relatively high fraction of the β-phase, are uniformly dispersed and can be simply eliminated in an acidic solution, which ensures that the PVDF-based film is not damaged [168].

TiO_2_ NPs are another typical example of an external filler that can enhance the mechanical property, β-phase proportion, and flexibility of electrospun PVDF nanofibers. For example, M. Alam et al. induced the β-phase in TiO_2_ NP-doped spin-coated PVDF nanocomposite (PNC) film [41]. TiO_2_ NPs can effectively enhance the overall performance of the NGs. However, the induction of metal oxide additives in the PVDF substrate is restricted by the agglomeration and conductive path development in the nano-additives with a high surface activity and large surface area, so considerable nano-scale metal oxide additives cannot be induced into PVDF. It can also reduce the dispersion of inorganic particles in the polymeric substrate [169]. To solve the problems of agglomeration and low percolation threshold, non-conductive SiO_2_ can be used to coat the NiO NPs before being added into the polymer. The agglomeration of NiO NPs can be reduced to a large extent with the SiO_2_ coating (Figure 5a) [170]. Pascariu et al. integrated graphene flakes and TiO_2_ NPs with the PVDF nanocomposite and obtained a fibrous membrane using electrospinning technology [171]. This novel nanocomposite showed obvious insulation properties at low frequency, in contrast to a conductivity above 500 MHz, which led to their application in electrostatic discharge (ESD) and electromagnetic shielding.

Carbon black (CB) (often 0–0.8 wt.%) has also been introduced as an additive in PVDF-TrFE/DMF solutions to fabricate PVDF-TrFE films with excellent piezoelectricity. Alamusi et al. investigated the effects of CB on the power generation capability of different prepared films by measuring the output voltages and harvested energy density [40]. When the CB content was 0.8 wt.%, the composite film was optimized with a calibrated open circuit voltage of 10.09 V, which was approximately 79% higher than that of the pure PVDF-TrFE films (5.63 V). In addition, the energy harvesting capability was increased by 164%. FTIR spectroscopy and other characterization techniques revealed that the addition of CB resulted in the preferential formation of the electroactive β-phase (with piezoelectricity) instead of the α-phase (without piezoelectricity), which was ascribed to the nucleate role of CB during the generation of composite films [40]. Mokhtari et al. compared the performance of various additives (ZnO, carbon nanotube (CNT), LiCl, and polyaniline) for selecting the optimum candidate for fabricating a flexible and lightweight NG [172]. Among these, the CNT NPs resulted in a higher content of β-phase PVDF and a higher output voltage (0.9 V) for electrospun web with a thickness of 230 μm.

I. Chinya et al. used electrostrictive polymer PVDF as the substrate and zinc ferrite (ZF) as a filler to prepare soft, flexible polymer–ceramic nanocomposite films [173]. The ZF filler was fabricated with a diameter of 50 nm using the sol–gel auto-combustion technique and encapsulated with polyethylene glycol-6000 (PEG-6000). The robust PEG layer acted as a coupling agent on the interface between the organic and inorganic phases and enhanced the Maxwell–Wagner–Sillars interfacial polarization by generating an interaction region with the Gouy–Chapman diffuse structure of polyglycolated ZF (Figure 5b). After surface modification, the induced polar phase and dielectric permittivity of the composites were improved. Concurrently, the dielectric loss significantly declined. Specifically, 10 wt.% ZF-PEG/PVDF exhibited the maximum polar phase of 92% with a maximum dielectric constant of 35 ± 5. Moreover, this modification enhanced the β-phase PVDF content in the composites, which led to an increase in the dielectric constant, energy storage density, energy discharge efficiency, and energy harvesting property of the nanocomposite [174,175,176,177,178,179,180,181,182,183,184,185]. Several modification techniques have been employed to improve the properties of nanocomposites. I. Chinaya and S. Sen enhanced the electroactive phase of PVDF by developing a nanocomposite with surface-modified ascorbic acid assisted phase pure zinc ferrite (ZF(ASC)), which was attributed to the presence of –SO_4_^2−^ and SiO_2_ clusters on the surface of the ZF [186]. In addition, the interfacial interaction between the ZF and -CF_2_ dipoles of the PVDF inside the nanocomposite was enhanced, which was conducive to the effective orientation of the PVDF dipoles [173].

In the ceramic-based piezoelectric materials, PZT is often used as a piezoelectric transducer. Compared with the polymer-based piezoelectric materials such as PVDF, ceramic-based piezoelectric materials have intrinsic drawbacks such as fragility and a lower piezoelectric voltage constant [150]. In addition, they are neither biocompatible nor environmentally friendly, due to the high lead content (~60%). This limitation can be avoided with several alternative ceramic-based lead-free piezoelectric materials such as BaTi_2_O_5_, KNbO_3_, and KNaNbO_3_. Among these, the non-perovskite metastable compound BaTi_2_O_5_ has excellent ferroelectricity, so it can be uniformly oriented in a PVDF polymer matrix to fabricate a piezoelectric energy harvester with large power generation capability. J. Fu et al. reported that the output power density of such a harvester reached 27.4 μW/cm^3^ across a load of 22 MΩ under an acceleration of 10 g [187]. Most significantly, the harvester exhibited good anti-fatigue performance and stability even after an extended period of cantilever vibration cycles. Most ceramic-based piezoelectric materials are ferroelectric with spontaneous electric polarization at a certain temperature, and the polarization direction can be varied by changing the external electric field. Similar to conventional perovskite ceramics, methylammonium lead iodide perovskite (MAPbI_3_) displays ferroelectric polarization and a change in structure, from tetrahedron to cube [188]. V. Jella et al. reported that the MAPbI_3_-PVDF composite films with 25 vol.% MAPbI_3_ exhibited a large dielectric constant of 56 at 1 kHz and remanent polarization of 0.83 μC/cm^2^ [189]. The special piezoelectric generator (PEG) showed a high open-circuit voltage (V_oc_) of 17.8 V and a short-circuit current density (J_sc_) of 2.1 μA/cm^2^, and the same PEG with an active layer thickness of 97.7 µm showed an enhanced V_oc_ of 45.6 V and J_sc_ of 4.7 μA/cm^2^. The ceramics contribute high piezoelectric coefficients and mechanical rigidity in composites, while PVDF offers flexibility and ease of processing. Additionally, blending ceramics with PVDF can provide enhanced electromechanical coupling, increased energy harvesting capabilities, and the potential for multifunctional device integration.

DNA is another nucleating agent for the nucleation of the β-crystallite phase in PVDF [57]. The negative charges on the surface of DNA produced by the phosphate backbone can lead to hydrogen bonding interaction with PVDF (Figure 5c). Furthermore, DNA is superior to hydrated salts and NPs, due to its nontoxic nature and biocompatibility. The DNA-mediated PVDF films exhibit ultrasensitive pressure response and dipole reversibility, due to the local orientation of the molecular dipoles.

Bio-inspired vitamin B_2_ (VB_2_) is considered to be an effective biocompatible alternative to non-toxic expensive stabilizers. S. K. Karan et al. introduced VB_2_ into an energy harvester for the first time as an effective stabilizer for β-phase PVDF (~93%). This harvester generated high output current (~12.2 μA) and voltage (~61.5 V). VB_2_ contains various hydroxyl groups, carbonyl groups, and amino groups in its backbone, which effectively stabilize the polar β-phases of PVDF through strong hydrogen bonding or electrostatic interaction with its –CH_2_–/–CF_2_– moieties (Figure 5d). A relevant working mechanism of PENG has been demonstrated according to the synergistic influence of PVDF dipoles (–CF_2_–/–CH_2_–) and the opposite surface charges on VB_2_ [190]. Positive and negative charge densities over the composite surface were obtained, which promoted the development of the piezoelectric polar β-phase via surface change polarization [156,164]. Furthermore, when external mechanical force is applied, the stress-induced polarization can lead to the directional alignment of PVDF dipoles [156,191]. Under the influence of pressure and surface charge-induced polarization, the molecular chains of PVDF can be self-polarized along a specific direction [190].

Several studies have focused on improving the electrical output capabilities of PVDF, as its piezoelectric coefficient (d_ij_) is much lower than that of ceramic materials. Recently, conjugated polymers have received considerable attention as promising candidate materials for piezoelectric NGs. Owing to their unique molecular structure and charge transfer properties, conjugated polymers with an excess of delocalized electrons are widely applied in organic electronics [192]. E. J. Ko et al. combined three conjugated polymers with a fullerene derivative called phenyl-C61-butyricacid methyl ester (PCBM_61_), an electron acceptor for creating blended systems which were then used to fabricate PVDF-based PENGs with an improved power generation capability [42].

Mokhtari et al. developed innovative triaxial braided PVDF yarn harvesters that transformed tensile mechanical energy into electricity via piezoelectricity [193]. Stretching or bending braided PVDF yarns generated a peak output voltage of 380 mV and a power density of 29.62 μWcm^−3^, which was approximately 1559% higher than the previously reported value for piezoelectric textiles. Furthermore, the developed triaxial NG showed obviously higher sensitivity than the PVDF-based NG. Unlike other piezoelectric harvesters, the triaxial braided PVDF yarn could be used for harvesting tensile energy, and exhibited outstanding reliability, which facilitated cycling with up to 50% strain for thousands of cycles without affecting the performance.

**Figure 5 polymers-15-02766-f005:**
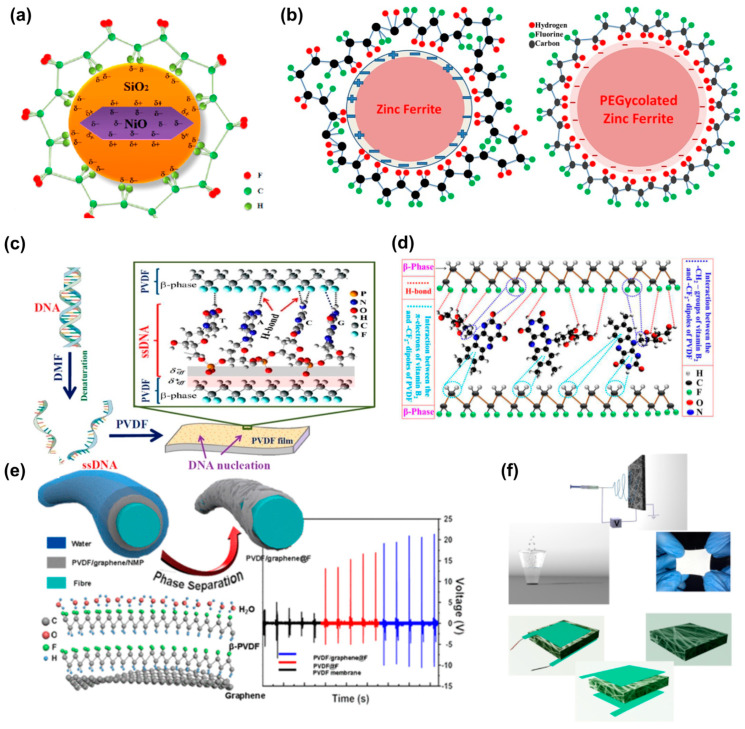
Schematic of the (**a**) β-phase formation process (reprinted from [170] with kind permission of ACS), (**b**) effects caused by the PEG layer (reprinted from [173] with kind permission of Elsevier), (**c**) DNA alteration in DMF, followed by the formation of DNA-PVDF film (reprinted from [57] with kind permission of ACS), (**d**) several interactions between PVDF dipoles and VB_2_ (reprinted from [190] with kind permission of Elsevier), (**e**) 3D structure models of β-PVDF formation (reprinted from [194] with kind permission of ACS), and (**f**) step-by-step fabrication of PENG (reprinted from [195] with kind permission of ACS).

Besides the aforementioned additives, many other materials can induce the β-phase or γ-phase transformation in PVDF, such as CNTs [31,36,37,51], carbon nanofibers (CNF) [134,196,197], and other inorganic materials [33,35,55,198,199,200,201]. Among these, inorganic filler is a big family, which can be divided into two categories: inert materials and active fast ionic conductors [16]. They are very hard and elastic, and provide binding sites for ion migration, thereby improving the mechanical properties of the composite system. It should be noted that some additives may be detrimental to the properties of PVDF film. For instance, the addition of more than 0.2 wt.% multi-walled carbon nanotube (MWCNT) may induce depolarization, to reduce the β-phase content [51].

Apart from the introduction of fillers or additives, structural modification is often applied in the composite materials. As shown in Figure 5e, the structural induction of graphene and water during the phase separation process results in the directional arrangement of the fundamental units of -CH_2_- and -CF_2_- of PVDF chains. Under optimized conditions, integrating the multi-layer PVDF/graphene composite onto fabric substrates results in a higher voltage output as compared to that of its film counterpart [194]. In addition, Maity et al. demonstrated a high-performance piezoorganic nanogenerator (PONG) based on the hybridization of sugar-encapsulated PVDF fabric nano-webs (Figure 5f) [195]. The presence of a sugar-interfaced structure caused a synergistic enhancement of piezoelectricity during the nanoconfinement of macromolecular PVDF chains. The fabricated PONG exhibited excellent output power density (up to ~100 V under 10 kPa human finger pressure and peak power density of 33 mW/m^2^) as well as sensitivity to abundantly available mechanical sources, including air flow, vibration, electrical devices, acoustic vibration, etc. Moreover, other organic or natural materials, especially bio-inspired natural piezoelectric materials, can be used for fabricating composite NGs because of their unique crystal structure, spontaneous piezoelectric property, easy accessibility, abundancy, cost-effectiveness, and outstanding biocompatibility.

Some researchers have combined the composite material fabrication with other methods to enhance the overall performance. For example, Khalifa et al. utilized the synergistic effect of electrospinning and nano alumina trihydrate (ATH) filler to enhance the content of the β-phase in PVDF. Different loadings of ATH were used as additives to fabricate the PVDF/ATH fabric nanocomposite [202]. The addition of ATH increased the surface charges of the electrospun droplets, resulting in thinner nanofibers. Moreover, the fraction of the β-phase reached 70.1% for the nanocomposite with 10% ATH. Another PENG was fabricated from a polyaniline (PANI)/halloysite nanotube (HNT)/PVDF blend nanocomposite by electrospinning [203]. HNT and PANi acted as a nucleating agent to enhance the fraction of β-phase PVDF, and PANI improved the electrical conductivity of the PVDF. Their synergism helped in improving the piezoelectric performance of the PVDF.

## 5. Structural Design

The electromechanical coupling coefficient of piezoelectric materials is influenced by several factors, such as the material structure, crystallite morphology, etc. When PVDF is employed as the main material to fabricate self-powered wearable electronic systems, the structural design is important. The structure of wearable sensors and human energy harvesters governs their performance and application. To obtain sufficient output power for practical applications, several novel structures have been proposed to improve the electromechanical coupling coefficient.

### 5.1. Topography of PVDF

The NFES technique has been utilized to fabricate different geometrical structures of PVDF fiber, including rods, tubes, particles, and flakes. Y. K Fuh et al. rearranged the parallel alignment of electrospun PVDF fibers into a concentric circle pattern, which made it possible to collect the mechanical energy when the deformation was along arbitrary directions [147,204]. As shown in Figure 6a,b, despite the change in topography and mechanical deformation direction, the output voltage and current reached 5 V and 400 mA, respectively [147,204]. Lee et al. utilized the rod-shaped P(VDF-TrFE) to fabricate a highly stretchable hybrid NG with a serration-like structure (Figure 6c) [205], boosting the crystallinity and energy harvesting efficiency. The novel arbitrarily directional PENG with concentrically circular topography can harvest more mechanical energy than a fiber-based generator with one-directional alignment [204]. Furthermore, the multi-directional structure of the generator allows it to harvest energy under multi-directional input forces [206].

Y. Mao et al. fabricated sponge-like mesoporous PVDF films with an enhanced β phase content [164]. These films were made by casting a mixture of PVDF solution and ZnO nanoparticles (NPs) on a flat surface, which was followed by HCl solution etching to remove the ZnO (Figure 6d). The average maximum open-circuit voltage and short-circuit current of the NG made with the PVDF film were approximately 11.0 V and 9.8 μA, respectively, where the supporting surface oscillated at 40 Hz. X. Chen et al. fabricated a novel self-connected, nanofiber-oriented vertically integrated P(VDF-TrFE) PENG using a patterned electrohydrodynamic (EHD) pulling technology [207]. The as-prepared NG showed a high output voltage of 4.0 V and a current of 2.6 μA, and the piezoelectric voltage was 5.4 times that of the bulk film. Similar vertically aligned PVDF fabric arrays have been obtained to manufacture NG by confined growth on a nanoporous substrate [208,209,210]. The piezoelectric effect of such arrays was enhanced by 1.85–3.40 times as compared to that of a traditional spin-coated film.

### 5.2. Multilayered Structures

The layered structure is the most common structure of film-based electrical devices (Figure 7a) [211]. M. H. Yung introduced a novel approach to enhance the piezoelectric output performance of PENG by using the layer-by-layer (LbL) method (Figure 7b,c) [212]. The PVDF-TrFE polymer film with piezoelectric properties and mechanical flexibility was utilized as the electroactive layer in the PENG. The maximum open-circuit voltage and closed-circuit current of the LbL multilayer PENG were 34 V and 100 nA, respectively [212]. The property of multilayer structure devices can be improved from another perspective, namely the tailoring approach. R. Guo et al. developed a self-powered dynamic monitoring sensor with a triple-layer strip structure (Figure 7d), showing distinct improvement in the output voltage, which could be optimized by varying the size [213]. To obtain sufficient deformation for generating power, C. Liu et al. prepared a PENG with self-amplified output by using micro-patterned polydimethylsiloxane (PDMS)/silver nanowires (Ag NWs)/PVDF sandwich structure (Figure 7e) [141]. The micro-patterned PDMS films provided excellent sensitivity to and output performance for the PENG. The corresponding maximum open-circuit voltage and peak short-circuit current were 1.2 V and 82 nA, respectively.

### 5.3. Arch Structures

Recently, arch structures have been combined with a multilayered structure to improve the NG performance [123,124,125,126,127,128,129,130,131,132,133,134,135,136,137,138,139,140,141,142,143,144,145,146,147,148,149,150,151,152,153,154,155,156,157,158,159,160,161,162,163,164,165,166,167,168,169,170,171,172,173,174,175,176,177,178,179,180,181,182,183,184,185,186,187,188,189,190,191,192,193,194,195,196,197,198,199,200,201,202,203,204,205,206,207,208,209,210,211,212,213,214,215,216,217]. The arch-shaped NG can scavenge energy from both the piezoelectric and triboelectric mechanisms [217]. For example, H. Fang et al. fabricated a novel arch-shaped TENG using a self-assembled polystyrene (PS) nanosphere array and a PVDF porous film, which exhibited an outstanding performance with an output voltage as high as 220 V per cycle [216]. The TENG is composed of two different nanostructures on every plate. The upper one is a PVDF porous layer fabricated using the spin coating process; the lower one is a polyethylene terephthalate (PET) film coated with a monolayer PS colloidal crystal. When a compressive stress is applied to the TENG, electrostatic equilibrium is removed temporarily. At the same time, the electric potential of the upper layer increases, due to the contact with the positively charged PS layer. Similarly, the lower electrode potential decreases. This causes an electron transfer, i.e., a charge flow between the upper and lower layers, due to the potential difference. Based on the same arch and multilayered structure, Cheng et al. designed a self-improving TENG (SI-TENG) by adding an internal plane-parallel capacitor (PPCS) [215]. In Figure 7f, Part I shows the SI-TENG with a friction layer composed of PVDF and polyamide-6 (PA-6) films, and Part II shows a PPCS, which contains two PET films with two electrodes separated by PVDF/epoxy resin layers on each film. In Part II, electrodes 1 and 2 and the covered PVDF/EP films form the PPCS. Part I is mainly used to generate energy under vibration. The charge generated by Part I is stored in the PPCS to form a high charge density in the device, which also improves the total output charge of the device.

### 5.4. Hybrid Structures

Besides the arch structures, some other structures have been proposed to simultaneously harvest triboelectric and piezoelectric energy [218]. Some hybrid structures can also harvest mechanical and solar energies [219,220] or both biochemical and mechanical energy [12], even including thermal energy [211]. L. Gao et al. demonstrated a double-helix multilayer structure to enhance the output performance of a TENG [221]. The double-helix-structured TENG consists of two parts (1 and 2). In Figure 7e, I and II show the cross-sectional views of parts 1 and 2, respectively. The negative electrode is composed of two attached PVDF films covered by copper foil (the positive electrode). Part 2 (the bottom electrode) shows a polytetrafluoroethylene (PTFE)-copper foil-PTFE structure, which is naturally separated from part 1 (the top electrode). All the layers are attached to each other without any external electrodes. When an external force is applied, an enhanced charge flow occurs from the bottom to the top electrode. Such a device can also be fabricated by an economical paper-folding process.

## 6. Application in Soft, Wearable Sensors and Energy Harvesters

### 6.1. Wearable Sensor for Exercise Monitoring

A PVDF sensor can be applied for the remote recognition of gestures in interactive human–machine interface systems [126,222]. E-skin can mimic the properties of human skin, and has received extensive attention due to its flexibility, stability, sensitivity, and biocompatibility [223]. For instance, W. Dong et al. developed a PVDF sensor as a human–machine interface in which the e-skin was used to monitor and classify signals, as well as to control the action of a remote robot [224]. Deng et al. designed a flexible self-powered PES based on cowpea-structured PVDF/ZnO nanofibers, and used it for remote gesture control in an interactive human–machine interface system [123]. Owing to the flexibility and the synergistic piezoelectricity of PVDF/ZnO, the piezoelectric sensor showed outstanding flexural sensitivity of 4.4 mV·deg^−1^ within the range of 44° to 122°, a short response time of 76 ms, and stable mechanical property.

Y.-K. Fuh and H.-C. Ho integrated printed circuit board (PCB)-technology-based self-powered sensors (PSSs) and direct-write NFES with PVDF micro/nano-fibers as source materials [224]. The piezoelectric sensors were assembled on gloves, bandages, and stockings to fabricate devices that could detect different kinds of human motions, including finger motion as well as flexing and extensions of an ankle. Chen et al. designed a highly sensitive sensor with a P(VDF-TrFE) nanowire array to detect finger motion, breathing, heartbeat, and low-magnitude sound waves [225]. This self-powered flexible sensor exhibited high sensitivity, good stability over 36,000 cycling tests, and excellent power generation performance. Furthermore, C. Dong et al. fabricated a flexible self-cleaning e-skin using PVDF/TiO_2_ nanofibers, where the nanofibers were synthesized using a high-voltage electrospinning technique. The e-skin exhibited a unique capacity to degrade organic pollutants, and could also monitor multiple body actions such as stressing, tensioning, finger bending, and fist clenching [144]. B. Dutta et al. demonstrated the high mechanosensing capability of a thin, flexible e-skin sensor based on NiO@SiO_2_/PVDF nanocomposites [170]. The e-skin sensor was highly sensitive, and could be used to accurately monitor the spatio-temporal distribution of stress stimuli in static and dynamic situations. Notably, it could classify the motion of different fingers [170].

### 6.2. Wearable Sensor for Health Monitoring

Besides e-skin, other kinds of soft, wearable sensors can be used to monitor healthcare and physiological conditions. Several PVDF-based sensors have been developed for detecting respiratory signals, body motion [144], muscular motions, and knee/elbow joint rehabilitation [125].

A respiratory monitor, also known as self-powered breath analyzer, plays a crucial role in medical diagnosis and treatment [216]. For example, it can be used to detect several internal diseases such as fatty liver disease (Figure 8a) [124]. To monitor heartbeat and respiration, S. Chen et al. developed a low-cost and highly sensitive non-contact system based on a flexible hollow-microstructure (HM)-enhanced self-powered pressure sensor [226]. Similarly, F. Wang et al. fabricated a PVDF piezopolymer film sensor for unconstrained in-sleep cardiorespiratory monitoring [14]. The sensor could detect respiration and heartbeats simultaneously by using wavelet multiresolution decomposition analysis. It should be noted that PVDF sensors do not need an extra energy supply, because of their self-powering property [224,225,227]. Therefore, PVDF sensors are extremely useful for monitoring human health conditions in real-time, in vitro and in vivo diagnostics, and smart electronics. I. Mahbub et al. proposed another continuous respiratory monitoring system based on pyroelectric material-based sensors (not just piezoelectric and triboelectric) [228]. The pyroelectric PVDF enabled the effective detection of breathing data with excellent sensitivity. Moreover, this system was convenient and harmless for neonatal infants having a chronic breathing disorder known as apnea of prematurity.

For another respiratory application, namely the removal of dust and fine particles, Liu et al. developed a novel self-powered electrostatic adsorption face mask (SEA-FM), based on the PVDF electrospun nanofiber film (PVDF-ESNF) and a TENG driven by respiration (R-TENG) [117]. The ultrafine particulates were electrostatically adsorbed by the PVDF-ESNF, and the R-TENG continuously provided electrostatic charges during this adsorption process by respiration. This mask exhibited a much better performance than commercial masks. Specifically, the removal efficiency of coarse and fine particulates was higher than 99.2 wt.%. Furthermore, the removal efficiency of ultrafine particulates was 86.9 wt.% after continuous use for 240 min and a 30-day interval.

C. Deng et al. developed a facile, self-powered insole plantar pressure mapping system (Figure 8b) [229] with a large pressure detection range using piezoelectric NGs and a self-designed data acquisition (DAQ) circuit board. The PENGs served as the sensor array for acquiring pressure signals and provided a wide detection range of pressure signals, and the circuit board was used to process and deliver the collected signals to a smartphone wirelessly by a program developed in Android. Remarkably, by combining with an electromagnetic TENG, a self-powered, continuous, and real-time pressure distribution monitoring system was obtained, which provided a viable option for acquiring sport/exercise biomechanics data, preventing injury, and predicting ulceration in the feet. M. Alam et al. designed another outstanding, self-powered integrated platform based on a PENG to generate electricity from the human gait. The acoustic sensitivity and energy conversion efficiency of this PENG reached 26 V/Pa and 61%, respectively [112].

Motivated by the structure and functions of the human fingertip, J. Park et al. fabricated fingerprint-like patterns and interlocked nanostructures in ferroelectric films (Figure 8c,d) [221]. The fingertip skin was composed of slow-adapting mechanoreceptors for static touch, fast-adapting mechanoreceptors for dynamic touch, free nerve endings for temperature, fingerprint patterns for texture, and epidermal/dermal interlocked nanostructures for tactile signal amplification. The skin could be used to simultaneously monitor the pulse pressure and temperature of artery vessels, precisely detect acoustic waves, and discern different surface textures. Furthermore, the ferroelectric e-skins could be used to monitor and distinguish different spatiotemporal tactile stimuli, such as static or dynamic touch, vibration, and temperature, with high sensitivities.

### 6.3. Soft, Wearable Energy Harvester

PVDF has been extensively applied for energy harvesting [13,230,231,232,233]. E. Häsler et al. were the first to realize the conversion of the energy expenditure for respiration into electric power [230]. They rolled two stacked PVDF sheets into a tube to fabricate the converter. The converter was fixed to the two adjacent ribs of a 25 kg mongrel dog to harvest its respiration energy. The maximum voltage of the harvester was 18 V, which corresponded to an output power of approximately 17 μW under continuous operation for 3 h. H. Zhang et al. conducted a similar experiment on a 30 kg porcine [231]. They developed a flexible and implantable PENG by using a flexible PVDF film coated with an aluminum layer. The PENG device with two copper electrodes was packaged with a polyimide (PI) membrane and wrapped around a latex tube. This device was wrapped around the ascending aorta of the porcine to harvest energy (Figure 9a). The maximum output voltage, current, and power of the PENG were 10.3 V, 400 nA and 681 nW, respectively. Until now, in vivo energy harvesting experiments have been usually limited to animals. In vivo human energy harvesting has been mainly attempted by simulation, such as using an artificial blood artery [13,233], and most relevant experiments have been performed in vitro, to harvest energy from human body motion or heat.

Several self-powered wearable electronic devices have been developed to convert human motions into electricity. X. Xue et al. fabricated a self-charging power cell (SCPC) (Figure 9b) using PVDF film as a separator [234]. This SCPC could convert footfall mechanical energy into electricity, and the output voltage was approximately 1.5 V. Using the direct-write NFES method to deposit piezoelectric PVDF nano/micro fibers on a PDMS substrate, Y.-K. Fuh et al. successfully fabricated a flexible and stretchable energy harvester [227]. The harvester was able to scavenge human motion in all directions. The harvester contained nearly 50,000 rows of well-aligned PVDF nano/micro fibers and could produce a maximum output voltage and current of up to 10 V and 40 nA, respectively. Utilizing PVDF-co-hexafluoropropylene (PVDF-HFP) as both a piezoelectric generator and a polymer matrix of a flexible capacitor, W. Tong et al. proposed an all-solid-state flexible generator–capacitor polymer composite film to convert low-frequency biomechanical energy into electric energy (Figure 9c,d) [235]. This device was able to convert the mechanical energy from finger movements into electrical energy, which was stored in situ.

Harvesting energy from multiple resources at the same time is also an interesting research direction. E. Kar et al. fabricated a PENG which could effectively harvest mechanical energy from human motions such as finger imparting, heel and toe motion, and wrist bending, as well as from the environment, such as wind and sound [236]. K. H. Oh et al. proposed a self-powered 3D activity inertial sensor (3DAIS), which could harvest energy through piezoelectric, electromagnetic, and triboelectric mechanisms from diverse energy sources such as human motion, 3D vibration, rotation, etc. [237]. A schematic of the 3DAIS is presented in Figure 9e. The device consists of a 3D printed spherical shell and multiple layers of tape, aluminum, PVDF, and PTFE films on the inner wall of the shell. The magnetic buckyball, encapsulated inside the shell, can move freely when the 3DAIS is in motion. The aluminum films act as the electrodes for external wire connections, while the piezoelectric PVDF film is sandwiched between the electrodes to form a capacitive structure. A layer of tape is placed between the shell and the aluminum electrode, which acts as a cushion to facilitate higher deformation of the PVDF film in the presence of a mass, thereby increasing the piezoelectric energy harvesting capability. The PTFE film, which is in contact with the buckyball, enables the triboelectrification effect under the motion of the magnetic buckyball inside the sphere. Besides the triboelectric energy harvesting, the movement of the magnetic buckyball also generates energy via electromagnetic induction in the wire coils wrapped around the shell’s surface. Therefore, the 3DAIS can simultaneously generate electrical output from the piezoelectric PVDF film, triboelectric surfaces, and electromagnetic coils, in the presence of mechanical motion.

**Figure 9 polymers-15-02766-f009:**
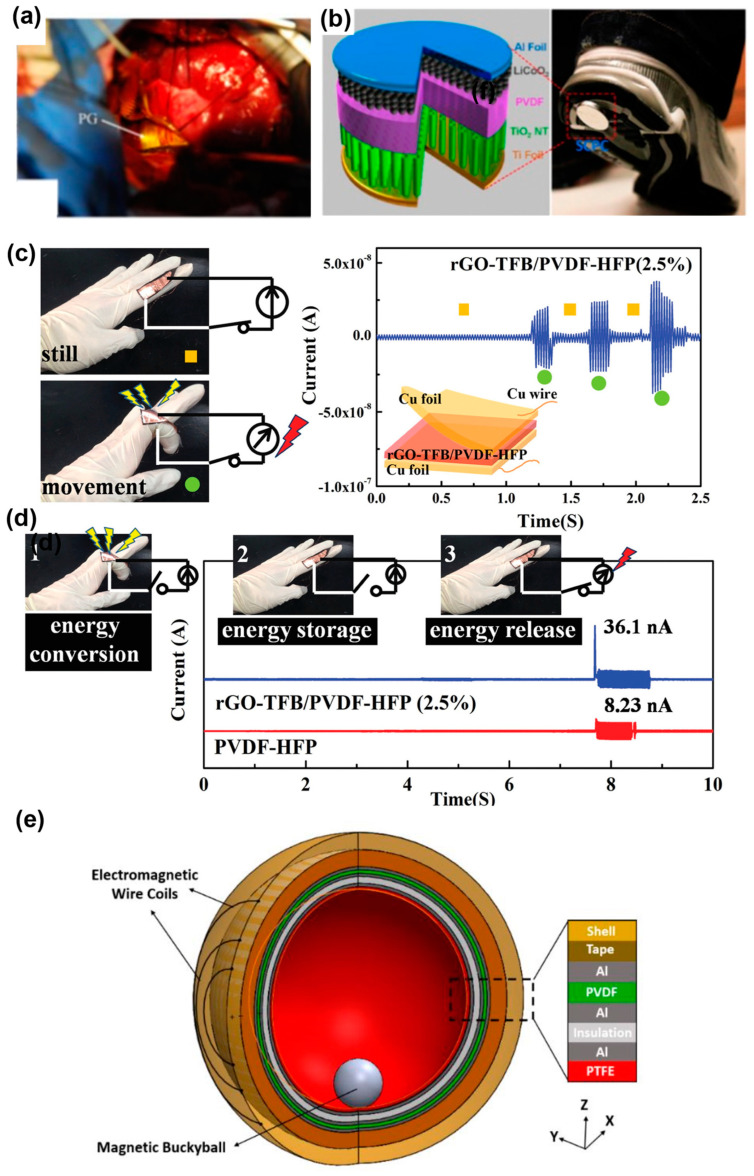
(**a**) Implantable PENG wrapped around the ascending aorta (reprinted from [231] with kind permission of Elsevier). (**b**) Structure of an SCPC with PVDF as a separator (reprinted from [234] with kind permission of ACS). (**c**) Harvester for generating energy from finger motion. (**d**) Output current vs. time curve of the PENG under finger bending (reprinted from [235] with kind permission of John Wiley and Sons). (**e**) Schematic illustration of the internal structure of 3DAIS (reprinted from [237] with kind permission of Elsevier).

## 7. Challenges and Perspectives

PVDF is often preferred in various devices, due to its excellent flexibility, but it has a limited sensitivity, due to the intrinsically low piezoelectric performance. The regulation of the ZnO-to-PVDF ratio is a promising approach for improving the sensitivity of the PVDF/ZnO-based piezoelectric sensor. W. Deng et al. designed a flexible self-powered piezoelectric sensor based on the cowpea-structured PVDF/ZnO nanofibers, which exhibited excellent bending sensitivity of 4.4 mV·deg^−1^, with a rapid response time of 76 ms, in a wide range, from 44° to 122° [123]. Among the different types of piezoelectric polymers, PVDF has a relatively lower electromechanical coupling factor (~0.30) and Curie temperature (~90 °C) [150,238]. The copolymer of PVDF, P(VDF-TrFE), has a higher electromechanical coupling factor than the individual materials. Therefore, it can be used as an alternative to PVDF in certain conditions. The relatively low Curie temperature of PVDF also limits the external applied environment, due to the reduced magnetic property at high temperatures, which can further affect the overall device performance. Meanwhile, modification of the film functionalities requires additional research. For example, it is necessary to improve and balance the piezoelectric and dielectric performance of PVDF films for specific applications, due to the individual limitations of some preparation methods [235]. Moreover, novel methods must be proposed to remove the opposing relationship.

Improving the performance and matching of electrode materials is another promising research direction. E. J. Ko et al. used nanofiber-type hydrophobic organic materials as electrodes for enhancing the performance of PVDF-based PENGs. The output signals (maximum voltages/currents) of PENGs (electrode/PVDF/electrode) were as follows: PENG-1 (PEDOT:PSS-CNT composite) 1.25 V/128.5 nA; PENG-2 (PEDOT-C4:DS) 1.54 V/166.0 nA; and PENG-3 (PEDOT-C6:DS) 1.49 V/159.0 nA. PENG-2 and PENG-3 showed an optimum piezoelectric output power of 63.0 nW and 59.9 nW, respectively, at 9 MΩ, which was 53.7% higher than that of PENG-1 (41.0 nW at 10 MΩ). The high output power was attributed to the excellent surface matching between the piezoelectric active material and the electrode materials [239].

For PVDF-based devices, energy storage and maintaining a stable output are always challenges. Usually, energy generation and energy storage are accomplished with two different units, so a part of the energy is inevitably wasted when the energy is transported from the generation unit to the storage unit. According to previous reports, the energy that can be harvested from human motion is relatively low [240]. At the same time, the maximum theoretical value of the electromechanical coupling coefficient is no more than 30% [230]. Therefore, it is crucial to reduce the energy wastage as much as possible. The energy wastage usually occurs in multilayered structures, which are fabricated using facile, convenient, and efficient processes. Xue et al. introduced a novel mechanism that directly hybridized the energy generation and energy storage processes into one for fabricating a self-charging power cell (SCPC) [234]. Specifically, the mechanical energy was directly converted and simultaneously stored as chemical energy without going through the intermediate step of first converting it into electricity. Comparison experiments proved that the single mechanical-to-chemical energy transformation technique for SCPC was far more productive than the twin mechanical-to-electric and electrical-to-chemical energy transformation procedures for conventional charging batteries.

Moreover, there are still some challenges for future research. Firstly, although electrospinning is a popular method for fabricating PVDF film, the direct-writing electrospinning is still regarded as a nascent but convenient technology for fabricating nano/microfibers [126]. Secondly, an effective composite model has not been proposed yet, such as combining the advantages of PENG and TENG. Some studies have utilized multiple sources for harvesting energy from piezoelectric, electromagnetic, and triboelectric mechanisms, which is a promising trend for future PVDF-based harvesters [237]. Thirdly, the optical transparency and structural flexibility issues need to be addressed for boosting the applications of self-powered devices [241]. In addition, it is necessary to continuously enhance the energy density and conversion efficiency for improving the overall device performance. Last but not least, as in the study of Crossley et al. [242], more figures of merit are necessary to be investigated comprehensively for wearable sensors and energy harvesters, due to the substantial PVDF-based composites and the application scenarios.

## 8. Summary

Owing to their several advantages, including biocompatibility, chemical resistance, film-forming ability, flexibility, stability, cost-effectiveness, and outstanding electroactive performance, including piezo-, pyro-, and ferro-electricity, PVDF and its copolymers exhibit immense application potential in several advanced fields, especially wearable sensors and human energy harvesters, which were highlighted in this review. Several structures and characteristics of the semicrystalline polymer PVDF were discussed, and the relevant phase identification techniques were described, including FTIR spectroscopy and XRD. Additionally, the methodologies for obtaining the electroactive phase of PVDF and its copolymers were discussed, including mechanical and temperature control, electric field poling and stretching, doping, and other unique methods such as the horizontal L-S technique. Additionally, direct methods for fabricating the electroactive phase, including electrospinning, melt spinning, solution blow spinning, and blending, were described. The advantages and disadvantages of different fabrication methods were comprehensively analyzed. Since the structure has a significant influence on the performance and application of PVDF-based devices, some structural designs were recommended as references. Furthermore, the application of the electroactive PVDF polymer in wearable sensors and human energy harvesters with various interesting functions in different conditions was explored. Finally, some existing challenges and perspectives were presented for boosting future research on the preparation, performance improvement, and application of PVDF-based electroactive polymers.

## Figures and Tables

**Figure 1 polymers-15-02766-f001:**
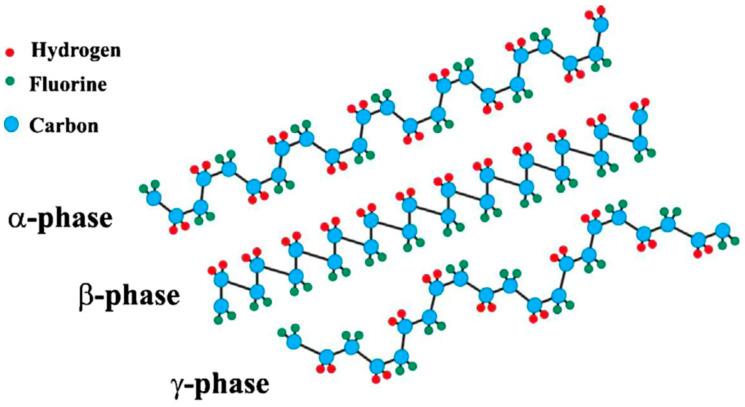
α-, β, and γ-phases of PVDF (reprinted from [47] with kind permission of Elsevier).

**Figure 2 polymers-15-02766-f002:**
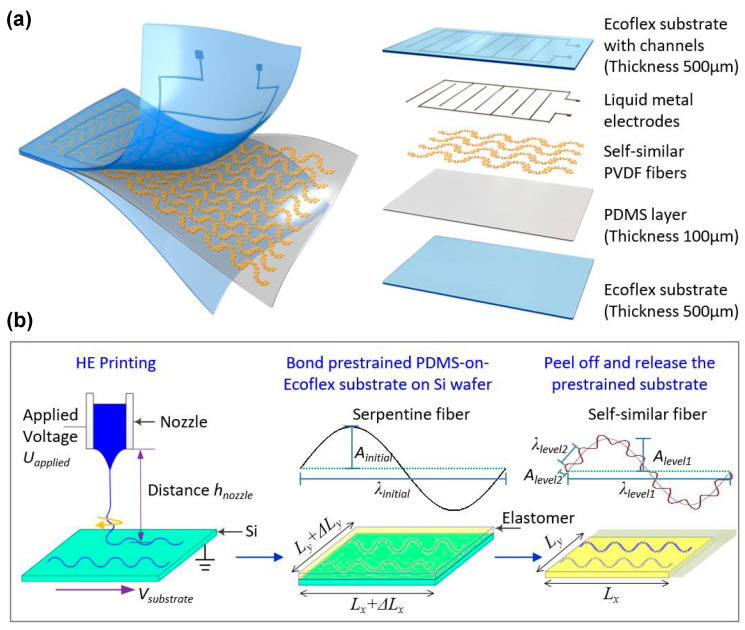
Schematic depiction of (**a**) multilayer films with self-similar PVDF fibers and combined layers, and (**b**) preparation route of self-similar nano/microfibers (reprinted from [93] with kind permission of Elsevier).

**Figure 3 polymers-15-02766-f003:**
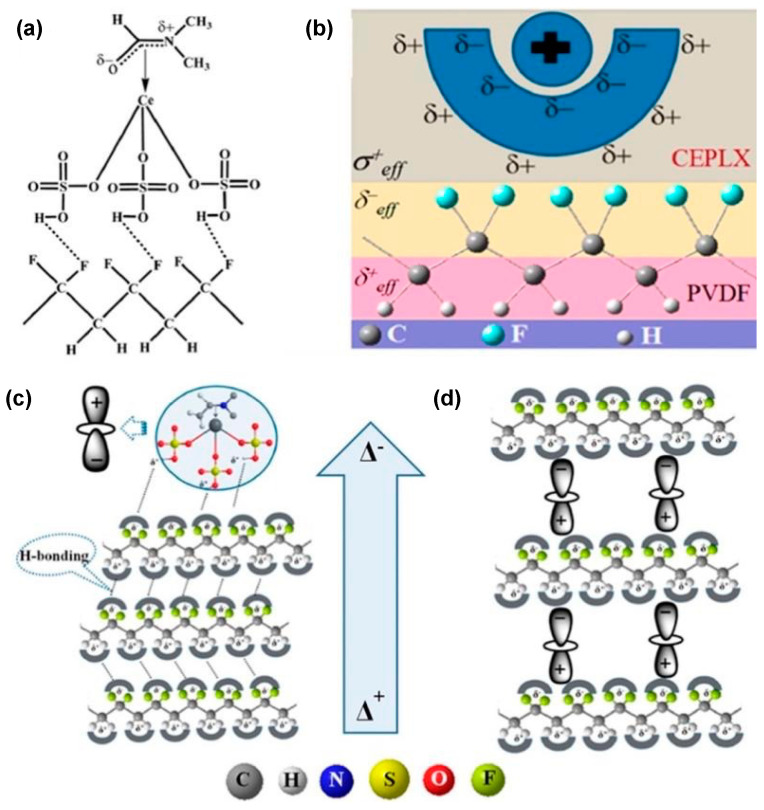
Schematic illustration of the (**a**) formation of hydrogen bonds between −HSO_4_ and CF_2_ dipoles, (**b**) electrostatic forces between the surface-active cation clusters of the cerium complex (CEPLX) and the CF_2_ dipoles. Self-poling due to the (**c**) formation of hydrogen bonds between molecules and (**d**) dipolar forces between the CEPLX and PVDF (reprinted from [114] with kind permission of ACS).

**Figure 4 polymers-15-02766-f004:**
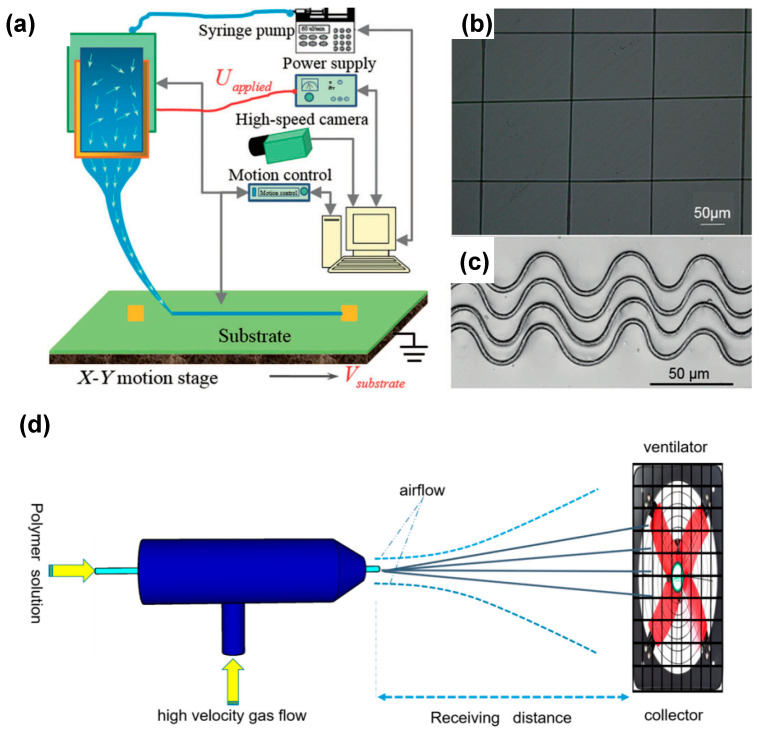
Schematic representation of (**a**) MES technique, (**b**) straight fibers written directly, (**c**) in-surface buckled fibers written directly (reprinted from [138] with kind permission of John Wiley and Sons), and (**d**) solution blow spinning (reprinted from [140] with kind permission of MDPI).

**Figure 6 polymers-15-02766-f006:**
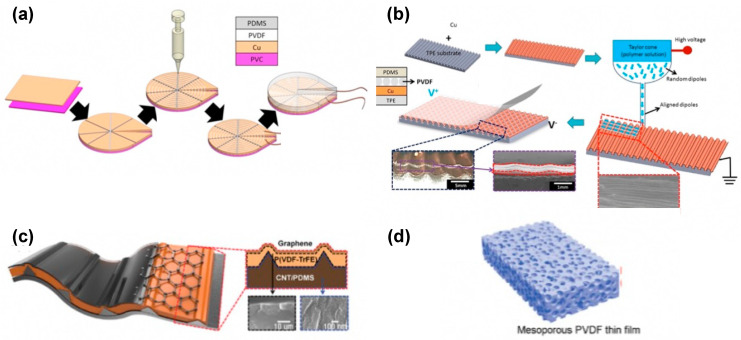
Schematic of the (**a**) NFES technique to directly write PVDF fibers with a concentric circle shape (reprinted from [204] with kind permission of Springer Nature), (**b**) NFES technique on wavy substrate (reprinted from [147] with kind permission of Springer Nature), (**c**) highly stretchable NG with a serration-like structure (reprinted from [205] with kind permission of John Wiley and Sons), and (**d**) structure of mesoporous PVDF thin films (reprinted from [164] with kind permission of John Wiley and Sons).

**Figure 7 polymers-15-02766-f007:**
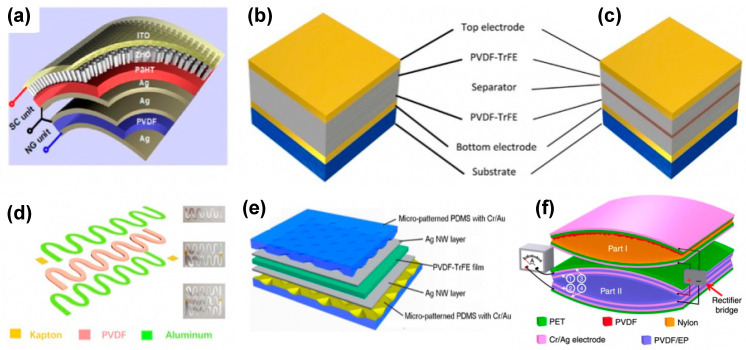
Schematics of the (**a**) prepared hybrid power cell (reprinted from [211] with kind permission of ACS), (**b**) and (**c**) stack-up LbL structure of PENG (reprinted from [212] with kind permission of Springer Nature), (**d**) strip and LbL structure of PENG and its physical photographs (reprinted from [213] with kind permission of Elsevier), (**e**) sandwich structure, also known as multilayered structure, of PENG (reprinted from [141] with kind permission of AIP Publishing), and (**f**) structure and operating mechanism of SI-TENG (reprinted from [214] with kind permission of Springer Nature).

**Figure 8 polymers-15-02766-f008:**
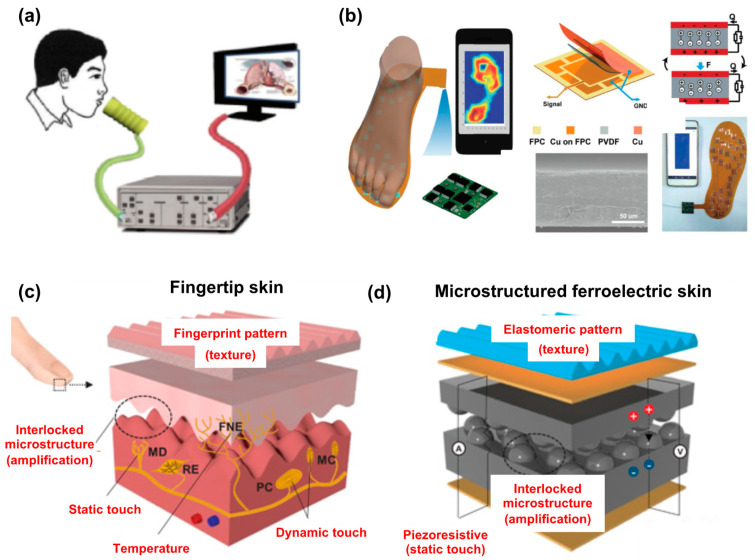
Schematic illustrations of (**a**) a breath analyzer for fatty liver diagnosis (reprinted from [124] with kind permission of Springer Nature), (**b**) foot pressure monitoring system and its functional principle, and SEM image of PVDF film (reprinted from [229] with kind permission of John Wiley and Sons). (**c**) Structure and working features of human fingertip skin, and (**d**) structure of fingerprint-like ferroelectric e-skin (reprinted from [218] with kind permission of AAAS).

**Table 1 polymers-15-02766-t001:** Characteristic FTIR absorption bands of α-, β-, γ-, and δ-phase PVDF.

Phase	Band Position (cm^−1^)	References
α	530	[48,49]
	615	[48,49,50]
	763–765	[48,49,50,51,52,53]
	795–797	[48,49,50,53]
	976	[49,51,53]
	1218	[54]
β	510	[48,49,55]
	836	[50]
	840	[47,48,51,55]
	845	[49]
	1210	[47,54]
	1274–1279 (shoulder)	[54,56]
	1383	[54,56]
	1423	[54]
	1431	[54]
γ	812	[47]
	833 (sharp)	[57]
	838 (broad)	[58,59]
	1233–1234 (shoulder)	[54,59]
δ	1182	[60]
	1209	[60]

**Table 2 polymers-15-02766-t002:** Diffraction angles and crystal planes of α-, β-, γ-, and δ-phase PVDF.

Phase	2θ (°)	Crystal Plane	References
α	17.6–17.7	(100)	[33,48,55,58,65]
	17.9	(110)	[66]
	18.68, 18.3–18.5,	(020)	[33,48,55,58,65,66]
	19.9	(021)	[65]
	19.9, 20.38	(110)	[33,48,55,58]
	20.2	(021)	[66]
	20.8	(011)	[33]
	26.5	(021)	[48,55,58]
	27.6, 25.6	(120)	[33,39]
	27.8,27.9	(111)	[65,66]
	35.7, 36.1	(200)	[65,66]
	39.0	(002)	[65,66]
	57.4	(022)	[65]
β	20.6–20.8	(110)/(200)	[33,48,55,65,66]
	36.3	(200)	[48]
	36.6	(020, 101)	[65,66]
	56.1, 56.9	(221)	[65,66]
γ	18.5	(020)	[55,65,66,67]
	19.2	(002)	[55]
	20.1–20.4	(110)	[55,65,66,67]
	26.8	(022)	[65,66]
	36.2	(200)	[58]
	38.7	(211)	[65,66]
δ	18.3	(020)	[39,68]
	17.6	(100)	[64]
	19.9	(110)	[39,68]
	25.6	(120)	[64]
	26.7	(021)	[39,68]
	28.1	(111)	[39,68]

## Data Availability

Not applicable.

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
