# Peer review of "The Preparation, Structural Design, and Application of Electroactive Poly(vinylidene fluoride)-Based Materials for Wearable Sensors and Human Energy Harvesters"

_polymers, 2023, doi:10.3390/polym15132766_

Round 1

Reviewer 1 Report

The manuscript “Preparation, Structural Design, and Application of Electroactive Poly(vinylidene fluoride)-Based Materials for Wearable Sensors and Human Energy Harvesters”, by Weiran Zhang, Guohua Wu, Hailan Zeng, Ziyu Li, Wei Wu, Haiyun Jiang, Weili Zhang, Ruomei Wu, Yiyang Huang, Zhiyong Lei, presents an up-to-date review concerning PVDF based materials for both sensors domain and human energy harvesters. The manuscript is well written, in general, having an important number of references. Also, the authors are kindly asked to undergo minor changes to the manuscript, to enhance the clarity and continuity of the present review. Please find my comments below.

a) Please insert the email of the corresponding author.

b) General remark: Please be aware when you insert the references. Please insert brackets to all references, otherwise could be misunderstood the text. For example, in Line 60, appears “PVDF 18”, which is quite wrong.

c)) Section 2.1 PVDF phase structure: the format of the reference numbers, for example, “28-37” is changed. Please be aware that inside the manuscript, the format of the text, references, etc., should be the same in all sections.

d)) Line 133, Equation 1: Please adjust the equation style format.

f) In Table 1, can identify the wavenumber, of the occurred vibrations. Is it possible to insert a graph with them? It should be more suggestive.

g) I suggest that Section 3, entitled “ Phase transformation methodologies” should be moved before Section 2 “PVDF phase structure and identification” because it is more clear to present how the phase was obtained, and after that to identify them with the FTIR and XRD. Please consider that first should be explained the principle, etc., and afterward, it can be exemplified by using dedicated analysis instruments.

h) Line 561: Please use the subscript for the chemical formula.

Author Response

Reviewer 1:

The manuscript “Preparation, Structural Design, and Application of Electroactive Poly(vinylidene fluoride)-Based Materials for Wearable Sensors and Human Energy Harvesters”, by Weiran Zhang, Guohua Wu, Hailan Zeng, Ziyu Li, Wei Wu, Haiyun Jiang, Weili Zhang, Ruomei Wu, Yiyang Huang, Zhiyong Lei, presents an up-to-date review concerning PVDF based materials for both sensors domain and human energy harvesters. The manuscript is well written, in general, having an important number of references. Also, the authors are kindly asked to undergo minor changes to the manuscript, to enhance the clarity and continuity of the present review. Please find my comments below.

Reply:

We thank Reviewer 1 for the positive comments and insightful suggestions on our manuscript. We have carefully revised our manuscript and provided point-by-point responses in the following pages.

Comments 1-a:

Please insert the email of the corresponding author.

Reply 1-a:

This is a critical comment. We have added the email address in the manuscript.

Comments 1-b:

General remark: Please be aware when you insert the references. Please insert brackets to all references, otherwise could be misunderstood the text. For example, in Line 60, appears “PVDF 18”, which is quite wrong.

Reply 1-b:

The comment is very important. We have revised this problem throughout the whole article.

Comments 1-c:

Section 2.1 PVDF phase structure: the format of the reference numbers, for example, “28-37” is changed. Please be aware that inside the manuscript, the format of the text, references, etc., should be the same in all sections.

Reply 1-c:

The comment is very important. We have revised this problem throughout the whole article.

Comments 1-d:

Line 133, Equation 1: Please adjust the equation style format.

Reply 1-d:

The comment is very important. We have revised this problem throughout the whole article.

Comments 1-f:

In Table 1, can identify the wavenumber, of the occurred vibrations. Is it possible to insert a graph

Reply 1-f:

This suggestion is excellent, and it can show the differences between various PVDF phases in FTIR spectra. Actually, we also considered adding a series of FTIR diagrams of different phases at the beginning. However, given the length of this review, the proportion of this section, and the focus, we omitted it eventually. Table 1 in the manuscript could be a reference for readers who are interested in it. The suggestion is valuable and it indicates the requirement for research in PVDF phases, so we recommend that this content would be included in the following paper in the future.

Comments 1-g:

I suggest that Section 3, entitled “Phase transformation methodologies” should be moved before Section 2 “PVDF phase structure and identification” because it is more clear to present how the phase was obtained, and after that to identify them with the FTIR and XRD. Please consider that first should be explained the principle, etc., and afterward, it can be exemplified by using dedicated analysis instruments.

Reply 1-g:

Thank you for your valuable suggestion regarding the section order. We truly appreciate your insightful comment and the time you have taken to do this. Your perspective on rearranging the sections is duly noted. After careful consideration, we have decided to maintain the current order as we consider it offers a logical flow and enhances reader understanding. We understand the importance of presenting the PVDF phase structure and identification (Section 2) before discussing the phase transformation methodologies (Section 3) to provide a solid foundation for readers. However, we will thoroughly re-evaluate the section order to ensure optimal clarity and coherence. We sincerely value your feedback and welcome any further suggestions or comments you may have, as your input greatly contributes to the improvement of our work.

Comments 1-h:

Line 561: Please use the subscript for the chemical formula.

Reply 1-h:

The comment is very important. We have revised this problem throughout the whole article.

Reviewer 2 Report

This review manuscript was well prepared for introduction to readers about the structural properties of PVDF, the pros and cons of its phases and structure designs for applications in wearable sensors and human energy harvesters. However, some issues were found in the review manuscript as mentioned in the below comments.  Thus, the reviewer suggests that this manuscript needs a minor revision before considering for publication.

Comment 1. The statement in line 50-54 of Introduction is unreliable and misleading about the drawbacks of PVDF in the fields of wearable sensors and energy harvesters. Ref [17] doesn’t support the mentioned drawbacks of PVDF. Ref 16 “suggested” some drawbacks for PVDF-based polymer batteries, which are not relevant to practical applications of PVDF in wearable sensors and energy harvesters.

Comment 2. The paragraph in line 55-71 does not explain reasonably why the PVDF-TrFE copolymer was created with higher crystallinity to replace the pristine PVDF in self-powered electronics and energy harvesting applications. Particularly, in the literature, PVDF-TrFE has higher crystallinity than PVDF [Ref: www.sciencedirect.com/topics/chemistry/polyvinylidene-fluoride], but the high crystallinity of PVDF is one of its drawbacks for practical applications [the author mentioned that in line 52-54]. So, why?

Comment 3. It is highly recommended to make Tables for comparing PVDF-based composites for sensors and TENG/energy harvesters.

Dear Editors,

Thank you for inviting me to check this review manuscript. 

Please see my comments for the authors. I suggest a minor revision for this manuscript. If you need my further help, please let me know!

Best regards,

LT Duy

Author Response

Reviewer 2:

This review manuscript was well prepared for introduction to readers about the structural properties of PVDF, the pros and cons of its phases and structure designs for applications in wearable sensors and human energy harvesters. However, some issues were found in the review manuscript as mentioned in the below comments. Thus, the reviewer suggests that this manuscript needs a minor revision before considering for publication.

Reply:

We thank Reviewer 2 for the positive comments and insightful suggestions on our manuscript. We have carefully revised our manuscript and provided point-by-point responses in the following pages.

Comments 2-1:

The statement in line 50-54 of Introduction is unreliable and misleading about the drawbacks of PVDF in the fields of wearable sensors and energy harvesters. Ref [17] doesn’t support the mentioned drawbacks of PVDF. Ref 16 “suggested” some drawbacks for PVDF-based polymer batteries, which are not relevant to practical applications of PVDF in wearable sensors and energy harvesters.

Reply 2-1:

This is a very important and meticulous comment. We have added some corresponding statements and changed the turn of Ref [16] and [17] to make the reference suitable.

Comments 2-2:

The paragraph in line 55-71 does not explain reasonably why the PVDF-TrFE copolymer was created with higher crystallinity to replace the pristine PVDF in self-powered electronics and energy harvesting applications. Particularly, in the literature, PVDF-TrFE has higher crystallinity than PVDF [Ref: www.sciencedirect.com/topics/chemistry/polyvinylidene-fluoride], but the high crystallinity of PVDF is one of its drawbacks for practical applications [the author mentioned that in line 52-54]. So, why?

Reply 2-2:

It is a very important comment. We have added a statement about why PVDF-TrFE has high crystallinity and why high crystallinity contributes to the electrical properties in this paragraph. Also, we revised the wrong statement of “high crystallinity of PVDF is one of its drawbacks”. Thanks for the review’s mention.

Comments 2-3:

It is highly recommended to make Tables for comparing PVDF-based composites for sensors and TENG/energy harvesters.

Reply 2-3:

This suggestion is excellent, and it can show the performance of different PVDF-based composites straightforwardly. Actually, we had considered making such a table to illustrate the influence of various materials blended in PVDF-based composites. However, when we paid attention to the different content of these materials mentioned in the literature, we realized it was hard to have a rigorous comparison among them. Moreover, we have carefully evaluated the revision due, so we recommend that the comparison be included in our following paper. This valuable comment gives us a novel perspective to investigate the differences in various PVDF-based composites.

Reviewer 3 Report

In the manuscript titled “Preparation, Structural Design, and Application of Electroactive Poly(vinylidene fluoride)-Based Materials for Wearable Sensors and Human Energy Harvesters” the authors review current status in the preparation of piezoelectrically active phases of PVDF and their use as sensors and energy harvesters. Topic of the paper is timely and interesting for the community, however, before its publication in journal Polymers comments and questions listed below should be addressed.

General comment. Structure is unclear and the manuscript is difficult to follow. Specifically, the authors split piezo-phase preparation techniques into “Phase transformation methodologies” and “Direct methods for preparing electroactive phases” chapters. But distinction between the two is unclear. Some techniques, i.e., adding oxide particles, appear in both chapters. Please re-structure them in a way that there is a clear distinction between the two (if you can’t then don’t split it) and avoid duplicating approaches in different parts of the manuscript.

Page 2, lines 52-54. The authors write: “However, they still have some drawbacks, such as low ionic conductivity, high crystallinity, and shortage of reactive groups, which limit their practical applications.” It is unclear why these properties are drawbacks. Be more specific.

Page 2, lines 64-66.  The authors write: “Further, the electromechanical coupling factor and piezoelectric coefficient of P(VDF-HFP) are much higher than those of PVDF and P(VDF-TrFE) copolymers.” It is to generic, please add numbers.

Page 2, lines 74-76. The authors write: “However, PVDF-based materials are more promising for wearable sensors and human energy harvesters due to their higher dielectric constant (10) and structural flexibility.” Figure of Merit (FoM) for energy harvesting is inversely proportional to dielectric permittivity therefore this statement is wrong.

Page 3, lines 94-96. The authors write: “Firstly, the α-phase, the most thermodynamically stable polymorph, is a non-electroactive, nonpolar, and paraelectric phase with low piezoelectricity and has a centrosymmetric (P21/c) monoclinic unit cell with alternating trans and gauge linkage (TGTG’) conformation.” If the structure is centrosymmetric its piezoelectric coefficient should be 0 and not “low”.

Page 3, line 114-116. The authors write: “The α-phase PVDF is the most stable phase of PVDF polymer, while the piezoelectricity of β- and γ-phases makes them electroactive.” This was already said, please omit.

Chapter 2. A table summarizing electromechanical properties of different phases would be very helpful to the reader. Also, discussion on the origin of negative d33 should be added.

Table 1 and Table 2: Exemplary XRD patterns and IR spectra would be very helpful to the reader.

Chapter 3 and 4. Stabilization of beta phase through geometrical confinement (https://onlinelibrary.wiley.com/doi/full/10.1002/aenm.201400519) should be added.

Chapter 3 and 4. It is difficult to draw any conclusions from these chapters. I suggest that the authors make a table where they summarize electromechanical properties obtained through different methods. They should focus on materials properties, not on devices (like power densities, etc., where device design also plays a major role).

Chapter 3.3. Title “Doping” is misleading. Doping is defined as adding an impurity, which modifies the electronic structure of the material. Here the authors discuss different fillers. Please change.

Page 13, lines 440-442. The authors write: “KNN and PZT (another popular piezoelectric ceramic material) are few lead-free ceramic based materials, which possess high-temperature stability and piezoelectric properties simultaneously.” PZT is an archetypal lead-based material. In addition, what do the authors mean by a word “few”?

Chapter 4.3. How does “blending” approach differentiate from “doping” in Chapter 3.3?

Page 15, lines 527-548. The authors discuss composites of piezoelectric ceramic powders with PVDF. However, d33 of ceramics is positive, while it is negative for PVDF. Can the authors comment on why this approach makes sense?

Chapter 6. Like before, it is very difficult to make conclusions from this Chapter. I suggest the authors to add: 1) Discussion on FoMs for sensors and energy harvesters; 2) A table where they summarize FoMs (and other relevant results) for all devices mentioned in the text.

English should be checked by a native speaker.

Author Response

Reviewer 3:

In the manuscript titled “Preparation, Structural Design, and Application of Electroactive Poly(vinylidene fluoride)-Based Materials for Wearable Sensors and Human Energy Harvesters” the authors review current status in the preparation of piezoelectrically active phases of PVDF and their use as sensors and energy harvesters. Topic of the paper is timely and interesting for the community, however, before its publication in journal Polymers comments and questions listed below should be addressed.

Reply:

We thank Reviewer 3 for the positive comments and insightful suggestions on our manuscript. We have carefully revised our manuscript and provided point-by-point responses in the following pages.

Comments 3-1:

General comment. Structure is unclear and the manuscript is difficult to follow. Specifically, the authors split piezo-phase preparation techniques into “Phase transformation methodologies” and “Direct methods for preparing electroactive phases” chapters. But distinction between the two is unclear. Some techniques, i.e., adding oxide particles, appear in both chapters. Please re-structure them in a way that there is a clear distinction between the two (if you can’t then don’t split it) and avoid duplicating approaches in different parts of the manuscript.

Reply 3-1:

Thank you for your valuable suggestion regarding the section order. We truly appreciate your insightful comment and the time you have taken to do this. The two sections are focused on synthesizing electroactive PVDF by transformation and single-step respectively. We also considered this issue in the manuscript preparation, so there are some distinguishing statements such as “An evident phase transformation from α- to β-phase in PVDF was obtained …...” and “This transformation is mainly attributed to the electrostatic interactions between the fluoride ions in PVDF……” in Section 3. According to this comment, we have evaluated carefully and deleted confusing content in Section 3 and 4 to make it more distinct. We sincerely value your feedback and welcome any further suggestions or comments you may have, as your input greatly contributes to the improvement of our work.

Comments 3-2:

Page 2, lines 52-54. The authors write: “However, they still have some drawbacks, such as low ionic conductivity, high crystallinity, and shortage of reactive groups, which limit their practical applications.” It is unclear why these properties are drawbacks. Be more specific.

Reply 3-2:

This is an important comment. We have revised the content more specific.

Comments 3-3:

Page 2, lines 64-66.  The authors write: “Further, the electromechanical coupling factor and piezoelectric coefficient of P(VDF-HFP) are much higher than those of PVDF and P(VDF-TrFE) copolymers.” It is to generic, please add numbers.

Reply 3-3:

It is a very important comment. The statement has been revised.

Comments 3-4:

Page 2, lines 74-76. The authors write: “However, PVDF-based materials are more promising for wearable sensors and human energy harvesters due to their higher dielectric constant (10) and structural flexibility.” Figure of Merit (FoM) for energy harvesting is inversely proportional to dielectric permittivity therefore this statement is wrong.

Reply 3-4:

This is an important comment. We have deleted the sentence for rigorous expression.

Comments 3-5:

Page 3, lines 94-96. The authors write: “Firstly, the α-phase, the most thermodynamically stable polymorph, is a non-electroactive, nonpolar, and paraelectric phase with low piezoelectricity and has a centrosymmetric (P21/c) monoclinic unit cell with alternating trans and gauge linkage (TGTG’) conformation.” If the structure is centrosymmetric its piezoelectric coefficient should be 0 and not “low”.

Reply 3-5:

It is a very important comment. The statement has been revised according to this comment.

Comments 3-6:

Page 3, line 114-116. The authors write: “The α-phase PVDF is the most stable phase of PVDF polymer, while the piezoelectricity of β- and γ-phases makes them electroactive.” This was already said, please omit.

Reply 3-6:

It is a very important comment. The sentence has been deleted according to this comment.

Comments 3-7:

Chapter 2. A table summarizing electromechanical properties of different phases would be very helpful to the reader. Also, discussion on the origin of negative d33 should be added.

Reply 3-7:

It is an important comment that can show the electromechanical properties clearly. According to this, we have added the contents of electromechanical properties and a discussion on the origin of negative d33 in Chapter 2. PVDF with different phases have distinct electromechanical coupling factors, and they also are influenced by temperature, poling condition, etc. thus, it is hard to show the specific data of the electromechanical properties, k. While this comment gives us a novel perspective to study further in the electromechanical properties under diverse conditions and we recommend to include it in our future work.

Comments 3-8:

Table 1 and Table 2: Exemplary XRD patterns and IR spectra would be very helpful to the reader.

Reply 3-8:

This suggestion is excellent, and it can show the differences of various PVDF phases in XRD and FTIR spectra. Actually, we also considered to add a series of XRD and FTIR diagram of different phases at the beginning. However, given the length of this review, proportion of this section, and the focus, we omitted it eventually. Table 1 and 2 in the manuscript has presented references for readers who are interested in it. The suggestion is valuable and it indicates the requirement for researches in PVDF phases, so we recommend that this content would be included in the following aper in the future.

Comments 3-9:

Chapter 3 and 4. Stabilization of beta phase through geometrical confinement(https://onlinelibrary.wiley.com/doi/full/10.1002/aenm.201400519) should be added.

Reply 3-9:

It is a very important comment. We have added the content and related reference in Section 3.4.

Comments 3-10:

Chapter 3 and 4. It is difficult to draw any conclusions from these chapters. I suggest that the authors make a table where they summarize electromechanical properties obtained through different methods. They should focus on materials properties, not on devices (like power densities, etc., where device design also plays a major role).

Reply 3-10:

This suggestion is excellent, and it can show the differences of electromechanical properties by different methods. Actually, we also considered to add such a table at the beginning. However, given the emphasis of different articles, it was hard to have a comprehensive comparison among them. Additionally, after we carefully evaluated the revision deadline and length of this review, we recommend maintain the current structure of this manuscript. We genuinely appreciate your feedback and eagerly invite any additional suggestions or comments you may have.

Comments 3-11:

Chapter 3.3. Title “Doping” is misleading. Doping is defined as adding an impurity, which modifies the electronic structure of the material. Here the authors discuss different fillers. Please change.

Reply 3-11:

It is a very important comment. The statement has been revised.

Comments 3-12:

Page 13, lines 440-442. The authors write: “KNN and PZT (another popular piezoelectric ceramic material) are few lead-free ceramic based materials, which possess high-temperature stability and piezoelectric properties simultaneously.” PZT is an archetypal lead-based material. In addition, what do the authors mean by a word “few”?

Reply 3-12:

It is a very important comment. The statement has been revised.

Comments 3-13:

Chapter 4.3. How does “blending” approach differentiate from “doping” in Chapter 3.3?

Reply 3-13:

Thank you for your valuable suggestion regarding the section distinction. Your perspective on rearranging the sections is duly noted. The two sections are focused on synthesizing electroactive PVDF by transformation and single-step respectively. We also considered this issue in the manuscript preparation, so there are some distinguishing statements such as “An evident phase transformation from α- to β-phase in PVDF was obtained …...” and “This transformation is mainly attributed to the electrostatic interactions between the fluoride ions in PVDF……” in Section 3. According to this comment, we have evaluated carefully and deleted confusing contents in Section 3 and 4 to make it more distinct. We sincerely value your feedback and welcome any further suggestions or comments you may have, as your input greatly contributes to the improvement of our work.

Comments 3-14:

Page 15, lines 527-548. The authors discuss composites of piezoelectric ceramic powders with PVDF. However, d33 of ceramics is positive, while it is negative for PVDF. Can the authors comment on why this approach makes sense?

Reply 3-14:

It is a very important comment. Blending ceramics with PVDF in the context of piezoelectric materials offers several advantages despite the contrasting piezoelectric constants. While it is true that ceramics typically exhibit a positive piezoelectric constant (d33), PVDF's negative d33 can still be advantageous in certain applications. By blending ceramics with PVDF, the resulting composite material can harness the combined benefits of both components. The ceramics contribute high piezoelectric coefficients and mechanical rigidity, while PVDF offers flexibility, ease of processing, and a broader operational temperature range. This combination allows for the development of flexible and versatile piezoelectric devices that can withstand varying conditions and exhibit improved sensitivity and durability. Additionally, blending ceramics with PVDF can provide enhanced electromechanical coupling, increased energy harvesting capabilities, and the potential for multifunctional device integration. Overall, this approach capitalizes on the unique strengths of each material to achieve a synergistic effect in piezoelectric applications. We have added the related statement in Section 4.3.

Comments 3-15:

Chapter 6. Like before, it is very difficult to make conclusions from this Chapter. I suggest the authors to add: 1) Discussion on FoMs for sensors and energy harvesters; 2) A table where they summarize FoMs (and other relevant results) for all devices mentioned in the text.

Reply 3-15:

This suggestion is excellent, and it can improve the understanding in the application of PVDF-based materials. While there is still a lack of sufficient study and data in this field because many factors can influence it, such as raw materials, driving scenarios, etc. Thus, we have added this valuable and promising perspective in Chapter 7 as a prospect point. Also, this could be a direction of our future research in the application of PVDF-based materials as sensors and energy harvesters. We genuinely appreciate the feedback you provide and eagerly invite any additional suggestions or comments you may have.

Comments 3-16:

English should be checked by a native speaker.

Reply 3-16:

This manuscript has been polished by native English speaker, and we provide the editorial certification as a PDF file.

Round 2

Reviewer 3 Report

The authors sufficiently adressed my comments, except the one on blending PVDF and ceramics. They write: "The ceramics contribute high piezoelectric coefficients and mechanical rigidity in composites, while PVDF offers flexibility, ease of processing, and a broader operational temperature range." Curie temperature of PVDF is slightly above 100 °C, while of PZT, for instance, it is ~400 °C. How can then PVDF offer broader operational temperature range? Please clarify of correct. 

Author Response

Comments from Review3: The authors sufficiently adressed my comments, except the one on blending PVDF and ceramics. They write: "The ceramics contribute high piezoelectric coefficients and mechanical rigidity in composites, while PVDF offers flexibility, ease of processing, and a broader operational temperature range." Curie temperature of PVDF is slightly above 100 °C, while of PZT, for instance, it is ~400 °C. How can then PVDF offer broader operational temperature range? Please clarify of correct. 

Reply: This comment is considerate and excellent. After carefully evaluating, we decide to delete the confusing statement about the " broader operational temperature range". We thank Review 3 for the positive comments and insightful suggestion on this manuscript.